# E3D-BENCH: A BENCHMARK FOR END-TO-END 3D GEOMETRIC FOUNDATION MODELS

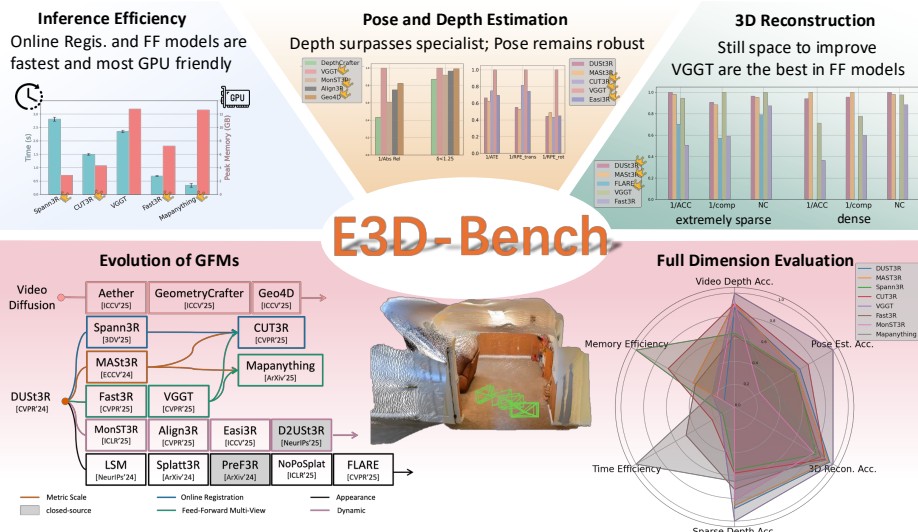

Figure 1: **E3D-Bench** evaluates 17 recent 3D Geometric Foundation Models (GFMs), spanning two major architectural families: feed-forward ViT-based and diffusion-based models, on both effectiveness and efficiency. For clarity, we visualize only the top methods per task. All metrics in bin chart except inference time, are averaged per scene, normalized, and converted to a "higher-is-better" scale for consistent comparison.

## ABSTRACT

Spatial intelligence, encompassing 3D reconstruction, perception, and reasoning, is fundamental to applications such as robotics, aerial imaging, and extended reality. A key enabler is the real-time, accurate estimation of core 3D attributes (camera parameters, point clouds, depth maps, and 3D point tracks) from unstructured or streaming imagery. Inspired by the success of large foundation models in language and 2D vision, a new class of end-to-end 3D geometric foundation models (GFMs) has emerged, directly predicting dense 3D representations in a single feed-forward pass, eliminating the need for slow or unavailable precomputed camera parameters. Since late 2023, the field has exploded with diverse variants, but systematic evaluation is lacking. In this work, we present the first comprehensive benchmark for 3D GFMs, covering five core tasks: sparse-view depth estimation, video depth estimation, 3D reconstruction, multi-view pose estimation, novel view synthesis, and spanning both standard and challenging out-of-distribution datasets. Our standardized toolkit automates dataset handling, evaluation protocols, and metric computation to ensure fair, reproducible comparisons. We evaluate 17 state-of-the-art GFMs, revealing their strengths and limitations across tasks and domains, and derive key insights to guide future model scaling and optimization. All code, evaluation scripts, and processed data will be publicly released to accelerate research in 3D spatial intelligence.

## 1 INTRODUCTION

3D reconstruction, perception, and reasoning are foundational steps in spatial intelligence, enabling downstream applications like robotic perception, extended reality, and embodied agents to interact

reliably with the physical world. A critical prerequisite is *fast*, *accurate* estimation of core 3D attributes (camera parameters, point clouds, depth maps, and tracked 3D points) from unstructured or streaming images. Traditional multi-stage pipelines—e.g., Structure-from-Motion (Schönberger et al., 2016; Yao et al., 2018; Gu et al., 2020) or optimization-based dense SLAM (Kerl et al., 2013; Whelan et al., 2016; Tateno et al., 2017)—incur high computational cost and maintenance complexity, further limiting their real-time deployment on latency-constrained scenarios such as robotic perception. They also struggle under sparse viewpoints, dynamic content, or non-ideal capture settings, leading to failures in open-world, latency-sensitive scenarios (Luo et al., 2020a; Lasinger et al., 2019).

**Rise of 3D Geometric Foundation Models (GFMs).** Driven by the success of large foundation models in language (Achiam et al., 2023) and 2D vision (Kirillov et al., 2023; Pagnoni et al., 2024), a new wave of generalizable 3D GFMs has emerged with end-to-end training and inference capabilities. These GFMs predict dense 3D representations (e.g., point maps and depth maps) directly from multi-view images, eliminating the need for slow-to-obtain or unavailable pre-computed camera parameters in open-world scenarios (Cong et al., 2025). Moreover, they unify feature extraction, matching, and geometry verification in an end-to-end single feed-forward pass with inference speeds in seconds, instead of traditional multi-stage reconstruction pipelines.

**Rapid Evolution of GFMs.** Since late 2023, this paradigm has seen explosive growth (See Tab. 1), reflecting a broader shift toward scalable, unified architectures for dense 3D understanding, empowering end-to-end *reconstruction* and *perception* in next-generation spatially aware systems. As illustrated in Fig. 1, recent GFMs adopt transformer-based architectures, and primarily follow two major architectural branches: ① **Feed-forward models** (e.g., DUSt3R (Wang et al., 2024c) and its variants (Leroy et al., 2024; Yang et al., 2025; Zhang et al., 2025a; Wang et al., 2025a; Fan et al., 2024b; Wang et al., 2025b)), which predict geometry in a single pass, and ② **Diffusion-based models** (e.g., Aether (Team et al., 2025), Geo4D (Jiang et al., 2025b)), which iteratively refine geometry through a denoising process. Notably, models derived from DUSt3R have diversified into memory-augmented designs, multi-view fusion strategies, and appearance-aware variants—each striking a unique balance among reconstruction quality, scalability, and inference efficiency to suit diverse deployment scenarios.

**A Timely Assessment of GFMs.** With the rapid proliferation of 3D GFMs, we ask:

Q ❶ Can GFMs serve as an *effective* and *robust* foundation for diverse 3D tasks and scenarios?

Q ❷ Can GFMs serve as an *efficient* foundation, especially for latency-constrained 3D applications?

These questions are critical because the final desired 3D GFMs for real-world 3D tasks must: ① Deliver high-quality 3D predictions, whether via end-to-end geometry output or through decomposed subtasks (e.g., depth and pose estimation). ② Generalize across diverse capture settings such as indoor vs. outdoor environments, sparse vs. dense view settings, ground-based vs. aerial perspectives, and static vs. dynamic scenes. ③ Able to operate under strict latency constraints, especially for real-time applications.

Hence, in the era of foundation models, we present the first systematic benchmark for 3D geometric foundation models, *aiming to uncover key trends that will guide future scaling and optimization.*

**First Comprehensive Benchmark for End-to-end 3D GFMs.** As summarized in Tab. 1, we evaluate a broad spectrum of GFMs, from pioneering feed-forward backbones (DUSt3R (Wang et al., 2024c), MASt3R (Leroy et al., 2024)) and their monocular-video adaptations (MonST3R (Zhang et al., 2025a)) to multi-view variants (VGGT (Wang et al., 2025a)) and diffusion-based models (Geo4D (Jiang et al., 2025b)), among others. *To rigorously assess **effectiveness***, we cover five core tasks: (a) Sparse-view depth estimation (b) Video-based depth estimation (c) Multi-view 3D reconstruction (both sparse and dense regimes) (d) Multi-view relative pose estimation (e) Novel view synthesis. *To probe **robustness***, we incorporate datasets spanning *diverse domains* beyond standard indoor scenes, including drone footage, dynamic real-world sequences, and air-ground paired views. Finally, we *benchmark latency and memory usage* to evaluate each model's readiness for latency-constrained, real-time deployment.

**Key Findings.** We summarize findings on 3D GFMs' effectiveness, robustness, and efficiency below:

    **1. Impact of Task Difficulty (§4.1)** : Current GFMs excel on simpler sub-tasks but struggle as complexity grows: ① Pair-view geometry inference outperforms true multi-view scenarios. ②

Table 1: **Overview of evaluated end-to-end 3D geometric foundation models.** Methods are grouped by input type. ✓ denotes support for metric scale; ✗ denotes normalized scale. FT indicates fine-tuning on a pretrained DUSt3R. Abbreviations are defined below. Confidence outputs are omitted for clarity. More comparisons of training configurations refer to Appendix Tab. S1.

| Method | ArXiv Date | Publication | Output Modality | Metric Scale | Multi-View Handling | Backbone | # Train Datasets | # Train Views | # Params |
|---|---|---|---|---|---|---|---|---|---|
| **Pair of Images** | | | | | | | | | |
| DUSt3R (Wang et al., 2024c) | 2312 | CVPR'24 | PM | ✗ | GA | ViT | 9 | 2 | 571.17 |
| MASt3R (Leroy et al., 2024) | 2406 | ECCV'24 | PM, M | ✓ | GA | ViT | 14 | 2 | 688.64 |
| LSM (Fan et al., 2024b) | 2410 | NeurIPS'24 | PM, 3DGS, S | ✗ | GA | ViT | 2 | 2 | 1100.07 |
| MonST3R (Zhang et al., 2025a) | 2410 | ICLR'25 | PM | ✗ | GA | ViT | FT 4 | 2 | 571.17 |
| NoPoSplat (Ye et al., 2024) | 2410 | ICLR'25 | 3DGS | ✗ | - | ViT | 1-2 | 2 | 625.27 |
| Align3R (Lu et al., 2025a) | 2412 | CVPR'25 | PM | ✗ | GA | ViT | FT 5 | 2 | 603.07 |
| Splatt3R (Smart et al., 2024) | 2408 | ArXiv'24 | PM, M, 3DGS | ✗ | GA | ViT | 1 | 2 | 744.79 |
| Easi3R (Chen et al., 2025c) | 2503 | ICCV'25 | PM | ✗ | GA | ViT | 0 | 2 | 571.17 |
| **Image Sequences** | | | | | | | | | |
| Spann3R (Wang & Agapito, 2025) | 2408 | 3DV'25 | PM | ✗ | OR | ViT | 6 | 5 | 658.69 |
| CUT3R (Wang et al., 2025b) | 2501 | CVPR'25 | PM, $PM_c$, Pose | ✓ | OR | ViT | 32 | [4, 64] | 793.31 |
| Aether (Team et al., 2025) | 2503 | ICCV'25 | PM | ✗ | FF | Diffusion | 1 | {17, 25, 33, 41} | 5571.76 |
| Geo4D (Jiang et al., 2025b) | 2504 | ICCV'25 | PM, Ray | ✗ | GA | Diffusion | 5 | 16 | 2610.09 |
| GeometryCrafter (Xu et al., 2025) | 2504 | ICCV'25 | PM | ✗ | GA | Diffusion | 14 | [1, 25]; [1, 110] | 2470.89 |
| **Multi-view Images** | | | | | | | | | |
| Fast3R (Yang et al., 2025) | 2501 | CVPR'25 | PM, $PM_c$, Pose | ✗ | FF | ViT | 6 | {20, 28} | 647.55 |
| VGGT (Wang et al., 2025a) | 2503 | CVPR'25 | PM, $PM_c$, Pose, M | ✗ | FF | ViT | 9 | [2, 24] | 1256.54 |
| MapAnything (Keetha et al., 2025) | 2509 | ArXiv'25 | $PM_c$, Pose, Scale | ✓ | FF | ViT | 13 | covisibility>25% | 563.34 |
| **Sparse-view Images** | | | | | | | | | |
| FLARE (Zhang et al., 2025b) | 2502 | CVPR'25 | PM, 3DGS, Pose | ✗ | FF | ViT | 8 | 8 | 1403.25 |

**Output Modality:** PM = Pointmap (world coordinate), $PM_c$ = Pointmap (camera coordinate), M = Matching, Pose = Camera Pose, Ray = Ray Map, 3DGS = 3D Gaussian Splatting, S = Semantics, Scale = Metric Scale Factor
**Multi-View Handling:** GA = Global Alignment, OR = Online Registration, FF = Feed-Forward, – = Not Supported
**Backbone:** All GFMs adopt transformer-based architectures. For clarity, ViT = Feed-Forward Transformer; Diffusion = Transformer-based Iterative Denoising
**# Params:** Measured in Millions (M). For diffusion models, we report parameters excluding the fixed VAE.

Single-attribute predictions (depth or pose) are more reliable than full 3D scene reconstruction. ③ Relative metrics (e.g., relative depth) are more accurate than absolute, metric-scale outputs.

2. **Impact of Data Domains (§4.2)** : GFMs generalize well on in-domain and out-of-domain data regimes, but degrade sharply under extreme distribution shifts, highlighting the need for more diverse training data.

3. **Insights on Model Architecture (§4.3)** : No single backbone type (feed-forward ViT or diffusion) dominates; architecture choice should align with task needs. Moreover, leveraging strong 2D feature extractors (e.g., DINO (Caron et al., 2021)) substantially boosts 3D performance.

4. **Efficiency Analysis (§4.4)** : Inference latency has improved, but modern GFMs still require tens of seconds to process hundreds of views, highlighting that efficiency—alongside accuracy—is essential for real-time deployment and for the future evolution of GFMs.

Looking ahead, despite promising progress, current GFMs still fall short of delivering a plug-and-play solution for end-to-end 3D attribute prediction in real-world spatial intelligence applications. Challenges such as metric-scale accuracy, generalization to diverse capture settings, and efficiency under latency constraints remain open. To facilitate progress in this direction, we present what is, to the best of our knowledge, the first systematic benchmark that spans a comprehensive set of 3D tasks and diverse real-world scenarios. We release all code and evaluation tools to support reproducibility and catalyze further research on robust, generalizable, and efficient 3D geometric foundation models.

## 2 BACKGROUND AND RELATED WORK

Due to space constraints, we provide only a brief overview of relevant work in Tab. 1. A comprehensive discussion of related literature is included in Appendix A.

## 3 BENCHMARK DESIGN AND RESULTS

To answer the core questions on the *effectiveness*, *robustness*, and *efficiency* of end-to-end 3D GFMs posed in Section 1, we select five representative tasks and assemble a diverse suite of datasets, ranging from standard indoor scenes to challenging scenarios such as drone-captured footage. Due to space constraints, each task table summarizes the dataset coverage. Full dataset details and preprocessing procedures are provided in Appendix Tab. S2.

## 3.1 SPARSE-VIEW DEPTH ESTIMATION

**Task:** *This task aims to predict the per-pixel depth maps, given a sparse view setting (views with minimal/no overlap).* This is a relatively underexplored task, with only a few recent end-to-end 3D GFMs, such as DUSt3R (Wang et al., 2024c) and Fast3R (Yang et al., 2025), explicitly adopting it in evaluation. However, *this task is critical for real-world deployment*, particularly in situations where sparse and unstructured image collections are the norm (e.g., internet photo collections of landmarks). In such cases, traditional pipelines like COLMAP (Schönberger & Frahm, 2016; Schönberger et al., 2016) often fail due to insufficient keypoint matches or poor overlap. Although most end-to-end GFMs are not explicitly trained for depth estimation, end-to-end GFMs naturally produce dense, stereo-consistent depth by projecting their predicted point maps and extracting the z-coordinate relative to each view. Assessing performance on sparse-view depth estimation, therefore, offers key insight into a model's ability to generalize and reason about 3D structure under severely limited visual input.

**Evaluation Protocol and Metrics:** Two widely used metrics for evaluation: Absolute Relative Error (AbsRel) measures the average relative error between model output and ground truth; the Inlier Ratio $\delta < 1.03$ (Uhrig et al., 2017; Eigen et al., 2014) captures the percentage of pixels within 3% relative error. We extract depth maps from the z-coordinate of predicted point maps, or use confidence-weighted averaging as in DUSt3R for views with multiple predicted depths. We test both normalized and metric-scale models, where we evaluate the former using median depth scaling, and the latter under their raw outputs and an extra median-aligned setting for fair comparison. All predictions are upsampled to full resolution before evaluation. We select quasi-optimal source views to reduce source view selection bias, following (Schroppel et al., 2022). More details can be found in Appendix C.1.

**Quantitative Results:** As shown in Tab. 2: 1) Most GFMs perform on par with or better than specialized baselines like Robust MVD (Schroppel et al., 2022), demonstrating strong robustness in this extreme case. 2) However, in the metric-scale setting, MASt3R, CUT3R, and Mapanything all fail to meet the strict threshold (e.g., $< 3\%$ error), highlighting the difficulty of accurate scale recovery under sparse inputs.

Table 2: **Comparison on Sparse-View Depth Estimation**. We report Absolute Relative Error (Abs Rel ↓) and $\delta<1.03$ accuracy (↑). Since LSM builds on top of DUSt3R without modifying its weights, their performance is nearly identical. For conciseness, we report them jointly as DUSt3R/LSM by default.

| Scale | Method | DTU | | ScanNet | | KITTI | | ETH3D | | T&T | |
|---|---|---|---|---|---|---|---|---|---|---|---|
| | | AbsRel ↓ | $\delta<1.03$ ↑ | AbsRel ↓ | $\delta<1.03$ ↑ | AbsRel ↓ | $\delta<1.03$ ↑ | AbsRel ↓ | $\delta<1.03$ ↑ | AbsRel ↓ | $\delta<1.03$ ↑ |
| | Robust MVD | 2.490 | 80.056 | 7.468 | 35.651 | 9.419 | 30.505 | 9.302 | 42.909 | 6.379 | 58.409 |
| Normalized | DUSt3R/LSM | 2.741 | 75.685 | 4.732 | 61.337 | 9.113 | 39.495 | 3.132 | 74.851 | 3.106 | 77.033 |
| | MASt3R | 3.343 | 68.301 | 5.949 | 54.516 | 9.542 | 46.805 | 2.471 | 81.291 | 2.381 | 82.262 |
| | Spann3R | 6.431 | 38.339 | 7.779 | 33.713 | 10.195 | 30.858 | 5.121 | 54.708 | 5.580 | 52.812 |
| | CUT3R | 6.200 | 47.421 | 8.231 | 39.464 | 23.849 | 12.087 | 5.224 | 59.864 | 4.594 | 56.773 |
| | VGGT | 1.085 | 94.305 | 4.386 | 64.968 | 9.436 | 41.309 | 1.782 | 86.337 | 2.075 | 85.174 |
| | Fast3R | 3.940 | 62.120 | 6.271 | 50.283 | 13.390 | 26.734 | 4.692 | 62.663 | 4.423 | 64.873 |
| | Mapanything | 9.072 | 57.107 | 9.180 | 46.394 | 26.477 | 10.901 | 6.505 | 62.239 | 8.062 | 63.265 |
| | MonST3R | 5.346 | 67.977 | 5.557 | 53.309 | 10.191 | 40.274 | 3.368 | 72.624 | 3.289 | 72.491 |
| | Robust MVD | 2.242 | 84.574 | 8.016 | 35.924 | 10.846 | 25.534 | 10.944 | 35.526 | 6.982 | 60.643 |
| Metric | MASt3R | 84.904 | 0.000 | 93.584 | 0.000 | 99.069 | 0.000 | 97.021 | 0.000 | 98.234 | 0.000 |
| | CUT3R | 84.904 | 0.000 | 93.584 | 0.000 | 99.069 | 0.000 | 97.022 | 0.000 | 98.234 | 0.000 |
| | Mapanything | 84.904 | 0.000 | 93.584 | 0.000 | 99.069 | 0.000 | 97.022 | 0.000 | 98.234 | 0.000 |

| | | | |
|---|---|---|---|
| **Gray** : Object-Centric | **Blue** : Indoor Scene | **Green** : Outdoor Scene | **Pink** : Mixed Scene |

## 3.2 VIDEO DEPTH ESTIMATION

**Task:** *This task evaluates producing temporally consistent depth maps from monocular video.* Coherent predictions across consecutive frames, despite motion blur, occlusions, or dynamic objects, are critical for applications such as robot navigation, SLAM, and AR. Methods like MonST3R (Zhang et al., 2025a) and Geo4D (Jiang et al., 2025b) are explicitly designed for dynamic scenes, but most existing GFMs have not been tested in temporally structured settings. We benchmark both dynamic-aware and static-scene models on video sequences, applying the latter to assess their zero-shot generalization to real-world dynamics. This enables analysis of not just single-frame accuracy but also the stability and robustness of depth predictions over time, key requirements for video-driven 3D perception systems.

**Evaluation Protocol and Metrics:** As in Sec. 3.1, we report *Absolute Relative Error* (AbsRel) and *Inlier Ratio* $\delta$ (Uhrig et al., 2017; Eigen et al., 2014) with a higher threshold 1.25, and evaluate both

normalized and metric-scale models. Models do not have access to camera intrinsics or ground-truth poses during inference. More details can be found in Appendix C.2.

**Quantitative Results:** As shown in Tab. 3: 1) Surprisingly, most GFMs outperform methods specifically designed for recent SOTA video depth estimation methods (e.g., DepthAnyVideo (Yang et al., 2024a), VideoDepthAnything (Chen et al., 2025a), DepthCrafter (Hu et al., 2024b), and Marigold (Ke et al., 2025)). 2) VGGT consistently achieves the best performance across all domains. 3) Geo4D is the next strongest overall, while other diffusion-based models perform less competitively. Among DUSt3R-style models, MonST3R and Align3R, both fine-tuned on video data, achieve the second best results. 4) Estimating depth in metric-scale remains more challenging; we observe significant improvements from MASt3R to CUT3R and Mapanything, reflecting better scale recovery and calibration with data and training scaling up. Visualizations could be found in Appendix D.1.

Table 3: **Comparison on Video Depth Estimation.** We report Abs Rel (↓) and $\delta<1.25$ (↑) cross diverse dataset domains.

| Scale | Method | Bonn | | TUM Dyn | | KITTI | | PointOdyssey | | Syndrone | | Sintel | |
|---|---|---|---|---|---|---|---|---|---|---|---|---|---|
| | | AbsRel ↓ | $\delta<1.25$ ↑ | AbsRel ↓ | $\delta<1.25$ ↑ | AbsRel ↓ | $\delta<1.25$ ↑ | AbsRel ↓ | $\delta<1.25$ ↑ | AbsRel ↓ | $\delta<1.25$ ↑ | AbsRel ↓ | $\delta<1.25$ ↑ |
| Normalized | DepthAnyVideo | 0.515 | 25.3 | 0.184 | 84.6 | 0.074 | 95.3 | 0.417 | 61.7 | 0.299 | 83.1 | 0.455 | 47.9 |
| | VideoDepthAnything | 0.268 | 48.3 | 1.101 | 89.0 | 0.060 | 98.2 | 0.283 | 70.3 | 0.138 | 92.5 | 1.691 | 45.4 |
| | DepthCrafter | 0.107 | 88.3 | 0.159 | 79.5 | 0.120 | 86.2 | 0.144 | 81.3 | 0.380 | 87.5 | 0.354 | 58.2 |
| | Marigold | 0.329 | 52.2 | 0.600 | 32.8 | 0.332 | 43.3 | 0.346 | 47.5 | 1.331 | 16.8 | 0.417 | 45.4 |
| | DUSt3R/LSM | 0.174 | 83.5 | 0.187 | 79.2 | 0.124 | 84.9 | 0.168 | 77.8 | 0.063 | 96.9 | 0.475 | 59.1 |
| | MASt3R | 0.160 | 81.5 | 0.162 | 83.1 | 0.082 | 93.2 | 0.150 | 79.3 | 0.046 | 97.5 | 0.374 | 63.9 |
| | Spann3R | 0.205 | 77.4 | 0.204 | 70.6 | 0.449 | 49.1 | 0.303 | 58.4 | 0.241 | 74.5 | 0.587 | 43.3 |
| | CUT3R | 0.068 | 95.0 | 0.108 | 84.7 | 0.104 | 89.9 | 0.095 | 88.4 | 0.111 | 89.5 | 0.466 | 56.0 |
| | VGGT | 0.056 | 96.3 | 0.068 | 93.9 | 0.051 | 96.6 | 0.026 | 99.0 | 0.075 | 95.9 | 0.242 | 65.9 |
| | Fast3R | 0.232 | 69.4 | 0.221 | 71.1 | 0.308 | 46.8 | 0.271 | 66.2 | 0.368 | 44.8 | 0.565 | 48.7 |
| | Mapanything | 0.105 | 91.0 | 0.168 | 85.1 | 0.223 | 80.7 | 0.172 | 80.6 | 0.212 | 79.4 | 0.456 | 51.8 |
| | MonST3R | 0.061 | 95.4 | 0.197 | 72.6 | 0.083 | 93.4 | 0.066 | 92.3 | 0.110 | 89.7 | 0.343 | 59.4 |
| | Align3R | 0.062 | 96.8 | 0.107 | 90.1 | 0.105 | 89.2 | 0.077 | 93.3 | 0.097 | 92.9 | 0.237 | 69.0 |
| | Easi3R | 0.061 | 95.8 | 0.192 | 76.9 | 0.150 | 76.2 | 0.143 | 82.1 | 0.095 | 94.0 | 0.323 | 53.9 |
| | Geo4D | 0.060 | 97.8 | 0.096 | 93.2 | 0.086 | 93.8 | 0.082 | 93.0 | 0.105 | 93.1 | 0.205 | 73.2 |
| | Aether | 0.582 | 61.2 | 0.192 | 80.6 | 0.065 | 96.2 | 0.123 | 87.9 | 0.145 | 91.1 | 0.343 | 69.4 |
| | GeometryCrafter | 0.061 | 96.8 | 0.115 | 87.7 | 0.410 | 53.8 | 0.124 | 83.6 | 0.123 | 90.8 | 0.280 | 72.4 |
| Metric | MASt3R | 0.536 | 3 | 0.445 | 35.7 | 0.627 | 1.1 | 0.518 | 4.1 | 0.935 | 0 | 0.682 | 21.5 |
| | CUT3R | 0.097 | 90.3 | 0.135 | 80.6 | 0.118 | 87.4 | 0.127 | 88.1 | 0.824 | 0 | 1.020 | 23.6 |
| | Mapanything | 0.347 | 22.3 | 0.334 | 52.0 | 0.270 | 75.6 | 0.248 | 63.9 | 0.292 | 46.0 | 0.895 | 26.0 |

**Blue** : Indoor Scene **Green** : Outdoor Scene **Orange** : Large Dynamic Motion **Cyan** : Drone Scene **Pink** : Mixed Scene

## 3.3 MULTI-VIEW RELATIVE POSE ESTIMATION

**Task:** *This task evaluates recovering relative camera poses from image collections,* a core capability for downstream tasks like localization, SLAM, and 3D mapping. Traditional SfM methods (Schönberger & Frahm, 2016; Schönberger et al., 2016) often struggle with sparse views or dynamic motion, but recent 3D GFMs show promise in estimating poses directly from raw images without explicit matching. Prior work has typically evaluated static models only on static datasets, and dynamic-aware models on a limited set of dynamic scenes. Our benchmark systematically tests all GFMs across both static and dynamic datasets to assess their generalization to diverse and challenging scenarios. We focus exclusively on relative pose estimation, as most GFMs operate up to an unknown global scale or in relative coordinate frames.

**Evaluation Protocol and Metrics:** We report three standard metrics: *Absolute Translation Error* (ATE), *Relative Translation Error* (RPE-trans), and *Relative Rotation Error* (RPE-rot), computed after applying Sim(3) Umeyama alignment (Umeyama, 1991) between predicted and ground-truth trajectories. For ULTRRA Challenge (Joshi et al., 2024), where a single Sim(3) alignment is infeasible since the aerial and ground trajectories are reconstructed in separate coordinate systems, we report alternative metrics to ATE in the Appendix D.2 and provide visualizations in Fig. S2.

**Quantitative Results:** As shown in Tab. 4: 1) Overall, multi-view VGGT and CUT3R, stereo-based methods (DUSt3R, MASt3R with test-time optimization), and diffusion-based Aether rank among the top-performing GFMs. 2) On long video sequences, such as ScanNet-eval, ADT, and TUM Dynamics, GFMs often exhibit increased rotation errors, while driving scenes from KITTI Odometry pose challenges in the form of high ATE. 3) Interestingly, despite being out-of-distribution, drone datasets like ACID and Syndrone are surprisingly well handled, indicating stronger generalization to aerial views than expected. 4) In contrast, GFMs perform significantly worse on the real-world ULTRRA challenge, which involves air-ground input pairs. A specialized method Aerialmegadepth (Vuong et al., 2025) combining DUSt3R with a diffusion backbone and domain-specific training shows improved performance, with per-dataset breakdowns in the Appendix D.2.

Table 4: **Evaluation on Multi-view Relative Pose Estimation.** We report ATE (↓), RPE translation (↓), and RPE rotation (↓) across six diverse scene types. ✗ indicates methods that are incompatible with pairwise inputs due to requiring a fixed number of input frames.

| Method | CO3Dv2 | | | ScanNet & ADT & TUM-Dyn. | | | KITTI Ordometry | | | Bonn & Sintel & Rel10k | | | ACID & Syndrone | | | ULTRRA | |
| | ATE | RPE$_{trans}$ | RPE$_{rot}$ | ATE | RPE$_{trans}$ | RPE$_{rot}$ | ATE | RPE$_{trans}$ | RPE$_{rot}$ | ATE | RPE$_{trans}$ | RPE$_{rot}$ | ATE | RPE$_{trans}$ | RPE$_{rot}$ | RPE$_{trans}$ | RPE$_{rot}$ |
|---|---|---|---|---|---|---|---|---|---|---|---|---|---|---|---|---|---|
| DUSt3R/LSM | 0.903 | 1.325 | 4.312 | 0.139 | 0.102 | 2.394 | 2.935 | 1.135 | 2.832 | 0.077 | 0.557 | 1.657 | 0.126 | 0.379 | 2.836 | 70.350 | 70.390 |
| MASt3R | 0.987 | 1.407 | 3.999 | 0.131 | 0.098 | 2.889 | 1.492 | 0.399 | 0.407 | 0.058 | 0.559 | 1.305 | 0.130 | 0.376 | 2.601 | 71.519 | 78.036 |
| Spann3R | 0.915 | 1.295 | 6.352 | 0.294 | 0.164 | 3.778 | 15.848 | 5.031 | 4.645 | 0.083 | 0.102 | 1.297 | 0.117 | 0.149 | 1.484 | 40.503 | 38.366 |
| CUT3R | 0.847 | 1.209 | 6.361 | 0.185 | 0.133 | 4.471 | 2.421 | 0.747 | 0.669 | 0.033 | 0.039 | 0.500 | 0.071 | 0.090 | 0.914 | 55.135 | 54.395 |
| VGGT | 0.478 | 0.704 | 2.264 | 0.113 | 0.086 | 1.535 | 0.955 | 0.315 | 0.335 | 0.062 | 0.111 | 0.580 | 0.280 | 0.461 | 0.802 | 63.451 | 77.281 |
| Fast3R | 0.698 | 1.035 | 4.352 | 0.499 | 0.391 | 23.739 | 22.109 | 7.573 | 7.366 | 0.111 | 0.170 | 2.017 | 0.436 | 0.518 | 1.979 | 51.149 | 54.150 |
| Mapanything | 1.616 | 2.371 | 13.102 | 0.183 | 0.134 | 3.087 | 3.415 | 1.373 | 0.817 | 0.059 | 0.108 | 0.596 | 0.277 | 0.592 | 2.812 | 58.459 | 88.158 |
| MonST3R | 2.456 | 3.327 | 23.458 | 0.448 | 0.286 | 12.817 | 2.426 | 0.782 | 0.949 | 0.098 | 0.152 | 0.830 | 0.335 | 0.504 | 1.514 | 70.388 | 77.325 |
| Align3R | 1.027 | 1.550 | 6.499 | 0.425 | 0.215 | 9.430 | 4.611 | 0.817 | 0.600 | 0.076 | 0.091 | 1.083 | 0.150 | 0.179 | 0.977 | 72.010 | 70.638 |
| Easi3R | 0.857 | 1.271 | 5.052 | 0.174 | 0.103 | 2.872 | 3.625 | 0.919 | 0.615 | 0.075 | 0.094 | 1.361 | 0.119 | 0.138 | 1.733 | 62.061 | 71.060 |
| Geo4D | 0.798 | 1.264 | 5.692 | 0.436 | 0.175 | 10.565 | 1.662 | 0.497 | 0.696 | 0.573 | 0.472 | 3.779 | 0.384 | 0.329 | 1.395 | ✗ | ✗ |
| Aether | 3.168 | 2.366 | 21.643 | 0.644 | 0.273 | 14.804 | 1.553 | 0.744 | 0.744 | 0.195 | 0.122 | 1.610 | 0.152 | 0.097 | 0.796 | ✗ | ✗ |

**Gray** : In Distribution  **Blue** : Long Sequence  **Green** : Street Driving  **Orange** : Indoor-Outdoor Scene  **Cyan** : Drone  **Purple** : Air-Ground

## 3.4 MULTI-VIEW 3D RECONSTRUCTION

**Task:** *This task assesses reconstructing dense 3D point clouds from multiple input views.* This evaluates GFMs for applications such as AR mapping, robot navigation, and neural rendering. Unlike traditional SfM pipelines like COLMAP (Schönberger & Frahm, 2016; Schönberger et al., 2016), which rely on keypoint matching and multi-stage optimization, end-to-end 3D GFMs can generate pointmaps directly in a feed-forward manner, even under sparse views or dynamic scenes where classical methods often fail. Recent studies show that sparse-view predictions from GFMs can be effectively combined with methods like 3D Gaussian Splatting (Fan et al., 2024a) for 3D reconstruction. However, prior work typically evaluates either sparse-view or dense-view settings alone. Our benchmark considers both, providing a more complete picture of each model's scalability, generalization, and reconstruction fidelity across varied capture conditions.

**Evaluation Metrics:** *accuracy* (Acc), the mean distance from predicted points to the ground truth, *completeness* (Comp), the mean distance from ground-truth to the predicted surface, and *normal consistency* (NC), the mean cosine similarity between predicted and ground-truth surface normals.

**Evaluation Protocol:** We evaluate: (1) **Extremely Sparse-view reconstruction** simulates real-world constraints with limited inputs, using 2–5 images per scene selected to have minimal or no overlap and wide baselines. (2) **Dense-view reconstruction** uses 10–50 images per scene, selected to ensure high coverage and significant view overlap. To ensure fair comparison, for models(e.g., Fast3R and CUT3R that generate both local and global coordinates, we only evaluate the global pointmap to test their direct inference capability. All models are evaluated without access to ground-truth camera parameters during inference. Predicted point clouds are aligned to ground truth using the Umeyama algorithm. Metrics are computed over valid regions, with official masks applied when available.

**Quantitative Results:** As shown in Tab. 5: 1) Among feed-forward methods, VGGT consistently performs best across settings. In sparse-view settings, CUT3R and FLARE also perform well, particularly on indoor scenes, suggesting they effectively leverage multi-view cues despite limited inputs. 2) Methods like MonST3R and Align3R underperform, and for simplicity, we report only MonST3R as a representative of this group. 3) COLMAP, while a standard baseline, is highly sensitive to hyperparameters and fails frequently under sparse inputs (Wang et al., 2024c; Schops et al., 2017b). Even under our dense-view setting, which is dense for GFMs but still sparse by traditional MVS standards, COLMAP fails completely on 21/21 DTU scenes and 10/11 TUM-RGBD scenes. On the few scenes where reconstruction completes, the results remain significantly less accurate and complete than those from GFMs. Refer to Appendix D.4 for visualizations.

## 3.5 NOVEL VIEW SYNTHESIS

**Task:** *This task evaluates synthesizing photorealistic novel views from a few input images, requiring both accurate geometry and appearance modeling.* We benchmark the subset of 3D GFMs with explicit appearance modeling, LSM (Fan et al., 2024b), NoPoSplat (Ye et al., 2024), Splatt3r (Smart et al., 2024), and FLARE (Zhang et al., 2025b). Unlike prior work that mainly focuses on in-domain data, our benchmark tests generalization across diverse cross-domain scenarios.

**Evaluation Protocol and Metrics:** As NoPoSplat does not support multi-view input, we adopt a 2-view input setting for all methods to ensure consistency. Models predict novel views from two source images. We report three standard metrics: *Peak Signal-to-Noise Ratio* (PSNR) (Hore & Ziou,

Table 5: **Comparison on Sparse-view and Dense-view 3D Reconstruction.** We report Accuracy (↓), Completeness (↓), and Normal Consistency (↑).

| Setting | Method | DTU | | | 7-Scenes | | | NRGBD | | | ScanNet | | | TUM-RGBD | | |
|---|---|---|---|---|---|---|---|---|---|---|---|---|---|---|---|---|
| | | ACC ↓ | Comp ↓ | NC ↑ | ACC ↓ | Comp ↓ | NC ↑ | ACC ↓ | Comp ↓ | NC ↑ | ACC ↓ | Comp ↓ | NC ↑ | ACC ↓ | Comp ↓ | NC ↑ |
| Extremely Sparse | DUSt3R/LSM | 1.731 | 1.936 | 0.786 | 0.146 | 0.181 | 0.744 | 0.144 | 0.154 | 0.867 | 0.474 | 0.420 | 0.714 | 1.108 | 0.746 | 0.724 |
| | MASt3R | 1.895 | 2.003 | 0.788 | 0.262 | 0.254 | 0.732 | 0.113 | 0.102 | 0.810 | 0.467 | 0.389 | 0.701 | 0.738 | 0.747 | 0.739 |
| | Spann3R | 6.275 | 5.460 | 0.705 | 0.189 | 0.188 | 0.653 | 0.255 | 0.262 | 0.628 | 0.487 | 0.408 | 0.617 | 1.561 | 1.002 | 0.621 |
| | FLARE | 3.406 | 3.950 | 0.491 | 0.152 | 0.154 | 0.704 | 0.060 | 0.056 | 0.839 | 0.357 | 0.302 | 0.561 | 0.515 | 0.486 | 0.677 |
| | CUT3R | 6.885 | 5.022 | 0.727 | 0.118 | 0.142 | 0.717 | 0.104 | 0.078 | 0.828 | 0.260 | 0.238 | 0.692 | 0.587 | 0.553 | 0.683 |
| | VGGT | 2.716 | 2.301 | 0.765 | 0.077 | 0.080 | 0.762 | 0.069 | 0.071 | 0.903 | 0.063 | 0.079 | 0.798 | 0.385 | 0.331 | 0.747 |
| | Fast3R | 4.493 | 3.681 | 0.735 | 0.149 | 0.116 | 0.692 | 0.361 | 0.201 | 0.782 | 0.546 | 0.306 | 0.621 | 0.955 | 0.630 | 0.627 |
| | Mapanything | 15.356 | 2.932 | 0.747 | 0.126 | 0.104 | 0.737 | 0.150 | 0.109 | 0.848 | 0.423 | 0.323 | 0.666 | 0.840 | 0.695 | 0.720 |
| | MonST3R | 20.145 | 10.322 | 0.603 | 0.276 | 0.277 | 0.677 | 0.471 | 0.458 | 0.659 | 0.623 | 0.541 | 0.594 | 1.688 | 1.031 | 0.670 |
| Dense | COLMAP | - | - | - | 6.101 | 4.268 | 0.552 | 10.587 | 8.733 | 0.561 | 19.127 | 11.362 | 0.515 | - | - | - |
| | DUSt3R/LSM | 1.284 | 1.349 | 0.720 | 0.022 | 0.029 | 0.709 | 0.035 | 0.024 | 0.838 | 0.026 | 0.025 | 0.784 | 0.620 | 0.474 | 0.718 |
| | MASt3R | 1.374 | 1.409 | 0.723 | 0.025 | 0.028 | 0.697 | 0.043 | 0.024 | 0.809 | 0.035 | 0.027 | 0.757 | 0.209 | 0.211 | 0.708 |
| | Spann3R | 6.505 | 3.110 | 0.668 | 0.176 | 0.087 | 0.599 | 0.343 | 0.073 | 0.661 | 0.262 | 0.118 | 0.606 | 0.635 | 0.930 | 0.662 |
| | CUT3R | 4.710 | 2.413 | 0.699 | 0.025 | 0.028 | 0.665 | 0.076 | 0.029 | 0.782 | 0.042 | 0.030 | 0.693 | 0.740 | 0.595 | 0.665 |
| | VGGT | 2.103 | 1.925 | 0.748 | 0.019 | 0.032 | 0.659 | 0.015 | 0.012 | 0.874 | 0.016 | 0.021 | 0.728 | 0.065 | 0.091 | 0.692 |
| | Fast3R | 3.647 | 2.319 | 0.695 | 0.046 | 0.057 | 0.636 | 0.059 | 0.028 | 0.772 | 0.200 | 0.077 | 0.711 | 0.625 | 0.337 | 0.610 |
| | Mapanything | 16.402 | 1.869 | 0.698 | 0.047 | 0.027 | 0.657 | 0.085 | 0.036 | 0.780 | 0.056 | 0.041 | 0.698 | 0.224 | 0.192 | 0.672 |
| | MonST3R | 14.455 | 7.508 | 0.636 | 0.100 | 0.091 | 0.648 | 0.336 | 0.246 | 0.665 | 0.346 | 0.293 | 0.599 | 1.138 | 0.948 | 0.591 |

**Gray** : Object-Centric      **Blue** : Indoor Scenes

2010), *Structural Similarity Index* (SSIM) (Wang et al., 2004), and *Learned Perceptual Image Patch Similarity* (LPIPS) (Zhang et al., 2018).

**Quantitative Results:** As shown in Tab. 6, 1) Models perform best on datasets they were trained on—for instance, NoPoSplat excels on ACID and RealEstate10k but drops significantly on DTU and ScanNet++. Similarly, FLARE performs best on ScanNet++, its training domain. 2) All models perform poorly on DTU, likely due to limited exposure to object-level geometry in training, which primarily includes scene-level data. 3) LSM consistently underperforms, which we attribute to errors in novel view pose optimization (Details in Appendix C.3). Since it rescales ground-truth poses using the ratio of ground-truth and predicted pointmap's scales, any misalignment in prediction can lead to inaccurate pose estimation and degraded results. Visualization is at Appendix Fig. S3.

Table 6: **Comparison on Novel View Synthesis**. We report PSNR (↑), SSIM (↑), and LPIPS (↓).

| Method | DTU | | | RealEstate10k | | | ScanNet++ | | | ACID | | |
|---|---|---|---|---|---|---|---|---|---|---|---|---|
| | PSNR ↑ | SSIM ↑ | LPIPS ↓ | PSNR ↑ | SSIM ↑ | LPIPS ↓ | PSNR ↑ | SSIM ↑ | LPIPS ↓ | PSNR ↑ | SSIM ↑ | LPIPS ↓ |
| LSM | 17.38 | 0.6274 | 0.3198 | 18.92 | 0.6677 | 0.3643 | 17.12 | 0.6860 | 0.3887 | 20.46 | 0.6160 | 0.3822 |
| NoPoSplat | 17.91 | 0.6306 | 0.2810 | 24.53 | 0.8450 | 0.1634 | 22.15 | 0.7988 | 0.2359 | 25.35 | 0.7774 | 0.1875 |
| FLARE | 17.01 | 0.5672 | 0.2901 | 22.15 | 0.7126 | 0.2363 | 23.19 | 0.8117 | 0.2201 | 22.44 | 0.6229 | 0.2818 |

**Gray** : Object-Centric      **Blue** : Indoor Scene      **Cyan** : Drone Scene

## 3.6 INFERENCE EFFICIENCY

**Task:** Inference efficiency determines the viability of 3D GFMs in latency-sensitive or resource-sensitive domains, e.g., interactive systems. We measure both runtime and peak memory usage: memory footprint indicates whether the model can run entirely on-device or must offload to slower external memory (e.g., off-chip DRAM), which incurs extra latency penalties (Zhu et al., 2024).

**Evaluation Protocol and Metrics:** For fairness, we test all models on the same hardware, a single 80GB NVIDIA A100 GPU. We vary the number of input views from 2 to 128 and report two metrics: *Inference time* per scene (seconds) and *Peak GPU Memory* (GB), averaged over 10 runs.

**Quantitative Results:** As shown in Fig. 2): 1) Global alignment (GA)-based GFMs (e.g., DUSt3R) incur significantly longer inference times and are more susceptible to out-of-memory (OOM) issues. Techniques such as window-based GA in MonST3R and sparse scene graphs in MASt3R help alleviate some overhead but do not fully resolve them. 2) In contrast, online registration methods (i.e., Spann3R, CUT3R) achieve significantly lower GPU memory usage due to their streaming architecture. Meanwhile, multi-view methods like Fast3R and MapAnything offer much faster inference, benefiting from their specific efficiency-oriented designs.

## 4 OUR FINDINGS

Beyond simply reporting results and highlighting the top-performing models for each task, we distill and share several high-level insights drawn from our comprehensive evaluation.

**Scope and Limitations of System-Level Benchmarking.** Before detailing our findings, we emphasize that our analysis relies on the inference-time evaluation of "off-the-shelf" GFMs. While this approach reflects the practical deployment reality for practitioners, it inevitably conflates architectural

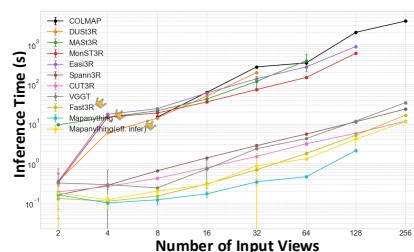 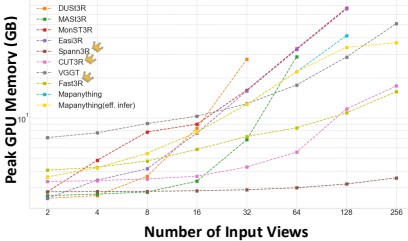

Figure 2: Inference Time (s) (Left) and Peak GPU Memory Usage (GB) (Right) on different numbers of views tested on a single 80GB A100. Time of COLMAP is tested on dense stereo with GPU.

differences with variations in training objectives, data scale, and auxiliary modules. We note that rigorously isolating these factors by retraining all models under a unified protocol is currently infeasible due to prohibitive computational costs (e.g., reproducing VGGT alone requires approx. $64\times$ A100 GPUs for over 9 days) and the unavailability of full training codes for several methods. Therefore, the trends discussed below should be interpreted as *observational characterizations* of the current state-of-the-art landscape rather than strictly causal architectural ablations. We refer readers to Appendix B and Tab. S1 for a detailed breakdown of the training data and configurations underlying each model.

## 4.1 WHAT IS THE IMPACT OF TASKS WITH DIFFERENT DIFFICULTIES?

**Multi-view geometry inference is inherently harder than pair-view inference.** GFMs behave fundamentally differently from classical pipelines like COLMAP. 1) Traditional SfM benefits from denser views via robust keypoint matching and triangulation (Schönberger & Frahm, 2016). However, GFMs perform end-to-end inference without such modules. Denser views bring challenges for global consistency. For instance, Tab. 5 shows that many GFMs struggle with dense multi-view reconstruction, and only a few (VGGT, DUSt3R, MASt3R) maintain consistency. 2) In addition, contrary to classical methods that often fail in sparse settings (Fan et al., 2024a; Wang et al., 2024c), GFMs like CUT3R and VGGT perform well under extreme view sparsity (Tab. 5), which is non-trivial.

**Directly predicting dense 3D scene representations is much more challenging than estimating individual 3D attributes like depth and camera poses.** Most GFMs achieve strong results on depth estimation, often outperforming task-specific baselines like DepthAnyVideo (Yang et al., 2024a). Their ability to predict complete 3D scenes remains limited, e.g., MonST3R ranks among the top-3 in video depth estimation (Tab.3) but performs poorly in 3D reconstruction (Tab.2), underscoring the increased difficulty of end-to-end global geometry prediction.

**Metric-scale depth estimation remains a key challenge for GFMs.** Unlike normalized depth, which is only consistent up to an unknown scale, metric depth requires accurate absolute scale, crucial for downstream tasks like localization and planning (Zhou & et al., 2017; Yang et al., 2020). However, predicting absolute scale is inherently more difficult than predicting relative ones. MASt3R fails consistently across settings (Tab. 3, 2), while CUT3R and MapAnything show progress but remain inconsistent: CUT3R struggles on drone data and sparse views, and MapAnything underperforms in non-drone scenes. Robust, domain-generalizable metric prediction remains an open problem.

**Global Alignment (GA) remains a strong post-processing step at the cost of efficiency.** In Tab. 5 on dense-view DTU, pure multi-view models (e.g., CUT3R, ACC↓ 4.710) underperform stereo methods with GA (e.g., DUSt3R, ACC↓ 1.284). GA aligns pair-wise predictions into multi-view consistent pointmaps and boosts geometric consistency. Extending pure feed-forward methods with GA could further improve the performance (also reflected in Table 1 in VGGT paper (Wang et al., 2025a)). However, GA incurs substantial latency, e.g., DUSt3R is 100× slower than VGGT on 32 views (Fig. 2), highlighting a key trade-off between accuracy and efficiency.

**Joint prediction of multiple geometric attributes (e.g., pose, depth, matching) may underlie recent performance gains.** Newer GFMs such as CUT3R, Geo4D, and Mapanything jointly predict pointmaps and camera poses, while VGGT further integrates a matching head for keypoint tracking. Such joint learning scheme encourages the network to learn richer and more structured representations that capture both spatial geometry and inter-view relationships, leading to improved generalization and robustness. Similar conclusions were also drawn in the original Geo4D and VGGT works.

*Takeaway ①: Current GFMs are promising but face significant challenges when learning from overly complex tasks.* **Recommendation:** Carefully decomposing difficult tasks (e.g., jointly predicting geometry, pose, depth, and tracking) into simpler sub-problems can facilitate more effective learning, especially under limited 3D data.

## 4.2 Do GFMs Generalize Well on Different Data Domains?

**GFMs struggle to generalize in domains with extreme data scarcity.** While GFMs show strong generalization to common out-of-distribution settings, such as aerial views, street-driving scenes, and egocentric perspectives, they often fail in extreme cases like large altitude variation or wide-baseline air-ground pairs, as seen in ULTRRA (Wang et al., 2024d). These failures are largely due to the absence of such examples in existing training data. As demonstrated in Vuong et al. (2025), introducing large-scale air-ground paired data substantially improves the performance of DUSt3R variants in these scenarios, confirming that domain generalization can be enhanced through targeted, diverse data. We include additional comparisons and analysis in the Appendix D.2.

This limitation also affects metric-scale depth estimation. Due to the scarcity of metrically accurate training data, GFMs often struggle to predict absolute depth. For instance, CUT3R and Mapanything, trained with broader supervision, outperforms MASt3R across multiple scenarios (Tab. 3).

*Takeaway ②: Diverse, high-quality data is critical for strong generalization.* To improve robustness in underrepresented domains, GFMs must be trained on data that covers broader distributions and metric-scale annotations.

## 4.3 Hints for Model Architecture Design, ViT or Diffusion? Strong 2D Feature Extractor?

**No single design, feed-forward ViT or diffusion, is universally superior.** While all current GFMs use a transformer-based architecture, they differ in execution: feed-forward ViTs process inputs in a single pass, while diffusion models iteratively refine predictions. Feed-forward designs offer greater flexibility in input configurations and scale more easily with data and modalities, making them well-suited for real-time and general-purpose applications. Diffusion-based GFMs, on the other hand, remain competitive in several tasks (e.g., Tab. 4 and Tab. 3). These observations suggest that backbone choice should depend on specific task demands and deployment requirements rather than a one-fits-all approach.

**Stronger 2D foundation models correlate with better 3D performance.** Modern 3D GFMs often rely on 2D backbones to extract visual features for 3D reasoning. We observe that utilizing robust 2D foundation models (e.g., DINOv2 Oquab et al. (2023)) correlates with substantially better generalization in 3D tasks. For instance, VGGT, which leverages a DINOv2 backbone, achieves state-of-the-art performance across nearly all evaluated tasks. Similarly, MapAnything utilizes DINOv2 to demonstrate robust capabilities in metric scale recovery and pose estimation in extreme out-of-distribution (OOD) scenarios. While our benchmark provides an observational, inference-only perspective, this finding is strongly corroborated by controlled experiments in the literature. Both VGGT (Wang et al., 2025a) and MapAnything (Keetha et al., 2025) report that switching to DINOv2 yields superior downstream performance, faster convergence, and greater training stability compared to a wide variety of alternative backbones, including CroCov2 Weinzaepfel et al. (2023), DUSt3R's image encoder Wang et al. (2024c), RADIO Heinrich et al. (2025); Ranzinger et al. (2024), and random initialization. This suggests that powerful 2D representations serve as critical priors for 3D GFMs, enabling 2D foundation model advancements to be effectively adapted for 3D perception.

**Multi-view aggregation strategies, ranging from post-processing to intrinsic architecture, dictate the trade-off between consistency and scalability.** As summarized in Table 1, we categorize multi-view handling into three mechanisms: Global Alignment (often a post-processing optimization adaptable to pairwise estimators), versus Online Registration and Direct Feed-Forward (intrinsic model designs tailored for specific data modalities).

- **Global Alignment (e.g., DUSt3R, MASt3R):** A *post-processing optimization* that aligns pairwise predictions into a unified scene graph. *Trade-off:* Achieves peak robustness on dense views (Tab. 5) via joint optimization. While naive implementations incur quadratic $O(N^2)$ costs (Fig. 2), recent

efficient designs employ sliding windows (e.g., MonST3R) or sparse matching graphs to alleviate scaling bottlenecks, though they remain computationally heavier than feed-forward approaches.

- **Online Registration (e.g., Spann3R, CUT3R):** An *intrinsic recurrent design* for streaming data that propagates geometry via latent memory. *Trade-off:* Enables linear $O(N)$ scaling and low peak memory for long sequences (Tabs. 3, 4), but often suffers from drift or lower global consistency on unordered image collections compared to global optimization.

- **Direct Feed-Forward (e.g., Fast3R, VGGT):** An *intrinsic attention-based design* that aggregates features from all views in a single pass. *Trade-off:* Offers the fastest, most scalable runtime and strong zero-shot generalization (Tab. 2), but often lacks the pixel-perfect consistency of optimization-based alignment in dense reconstruction tasks (Tab. 5).

*Takeaway ③: No single backbone (ViT vs. Diffusion) dominates. Architecture choice should align with task needs, where strong 2D priors (e.g., DINO) correlate with better generalization. Furthermore, the multi-view aggregation strategy, ranging from Global Alignment to Direct Feed-Forward, fundamentally dictates the trade-off between geometric consistency and inference scalability.*

### 4.4 ARE CURRENT GFMs READY FOR REAL-TIME PERCEPTION SYSTEMS?

**Despite notable progress, current GFMs are not yet ready for real-time deployment.** Real-time 3D perception systems require spatial encoders capable of fast and efficient inference. Although recent GFMs, particularly feed-forward models like Fast3R and MapAnything, have incorporated design optimizations to improve latency, none achieve true real-time performance. For example, even the most efficient models still require tens of seconds to process 256 input views (Fig. 2), making them unsuitable for time-critical applications such as robotics or AR.

*Takeaway ④: As GFMs scale to handle more views and complex tasks, efficiency becomes as critical as accuracy for enabling real-time 3D perception.*

## 5 CONCLUSION AND LIMITATIONS

We present the first comprehensive benchmark evaluating 17 recent end-to-end 3D Geometric Foundation Models (GFMs) across five core tasks, aiming to rigorously assess their effectiveness and efficiency. Our study reveals key insights into the impact of task difficulty, domain generalization, architectural design, and inference efficiency. While GFMs demonstrate strong potential for unified and scalable 3D perception, several challenges remain—particularly in metric-scale depth prediction, sparse-view reconstruction, and real-time deployment.

A limitation of our benchmark is that, in keeping with prior GFM studies, all evaluations use GPU setups. Future efforts should explore scaling GFMs through efficient inference techniques such as quantization and sparsity, and extend our evaluations to edge and mobile devices for low-resource deployment by reusing our benchmarks.

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

## .1 CLARIFICATION OF LLM USAGE

In this work, we employ LLMs to polish the writing throughout the paper and to assist in generating code for figure plotting.

# A MORE RELATED WORKS

## A.1 END-TO-END 3D RECONSTRUCTION

In this work, we define end-to-end 3D reconstruction as the task of *using fully differentiable models to directly map raw pixels from multiple images into pixel-aligned 3D point maps, without requiring explicit camera intrinsics or extrinsics at inference time.* End-to-end 3D geometric foundation models (GFMs)(Lu et al., 2024), which enable dense 3D perception directly from raw images, have recently emerged as powerful alternatives to traditional multi-stage pipelines(Schönberger & Frahm, 2016; Schönberger et al., 2016). Pioneering work such as DUSt3R (Wang et al., 2024c), pretrained on large-scale image correspondences (Weinzaepfel, Philippe and Leroy, Vincent and Lucas, Thomas and Brégier, Romain and Cabon, Yohann and Arora, Vaibhav and Antsfeld, Leonid and Chidlovskii, Boris and Csurka, Gabriela and Revaud Jérôme, 2022; Weinzaepfel et al., 2023), predicts dense pointmaps from stereo pairs and implicitly estimates correspondences—achieving state-of-the-art performance in depth and pose estimation.

**Stream 1: Feed-forward GFMs extending DUSt3R.** A major branch of GFMs evolves from DUSt3R, extending it across multiple axes, including metric-scale prediction, dynamic scene modeling, and multi-view inference. For instance, MASt3R (Leroy et al., 2024) adds metric depth prediction with enhanced matching accuracy. To address dynamic content, MonST3R (Zhang et al., 2025a) and Align3R (Lu et al., 2025a) incorporate task-specific fine-tuning, while Easi3R (Chen et al., 2025c) enables dynamic reconstruction without auxiliary cues. Other models such as Stereo4D (Jin et al., 2025), Uni4D (Yao et al., 2025), ZeroShot-MSF (Liang et al., 2025), Dynamic Point Maps (Sucar et al., 2025a), D$^2$USt3R (Han et al., 2025), MMP (Park & Shin, 2025), POMATO (Zhang et al., 2025c), BA-Track (Chen et al., 2025b), STORM (Yang et al., 2024b), and DPV (Sucar et al., 2025b) jointly model motion and geometry. To move beyond stereo pairs, Spann3R (Wang & Agapito, 2025), CUT3R (Wang et al., 2025b), and Regist3R (Liu et al., 2025b) introduce memory and recurrence for longer sequences, while Fast3R (Yang et al., 2025), VGGT (Wang et al., 2025a), MASt3R-SfM (Duisterhof et al., 2024), MUSt3R (Cabon et al., 2025), Light3R-SfM (Elflein et al., 2025), MV-DUSt3R+ (Tang et al., 2025), and Mapanything (Keetha et al., 2025) support scalable multi-view inference with outputs in global coordinates. Several of these models also support appearance modeling for view synthesis. Methods like LSM (Fan et al., 2024b), FLARE (Zhang et al., 2025b), Splatt3R (Smart et al., 2024), NoPoSplat (Ye et al., 2024), SpatialSplat (Sheng et al., 2025), Matrix3D (Lu et al., 2025b), AnySplat (Jiang et al., 2025a), VicaSplat (Li et al., 2025e), and PREF3R (Chen et al., 2024b) predict 3D Gaussians for high-quality rendering. Notably, LSM further enables language-guided semantic understanding, such as text-driven 3D semantic segmentation. Recent works also explore how to incorporate more available input conditions (Jang et al., 2025; Keetha et al., 2025), and further extend GFMs to broader tasks such as surface reconstruction (Raj et al., 2024), plane reconstruction (Liu et al., 2025a), SLAM (Li et al., 2025d; Maggio et al., 2025; Murai et al., 2025; Zheng et al., 2025; Liu et al., 2025c; Li et al., 2025a), end-to-end aerial-ground 3D mapping (Vuong et al., 2025), autonomous driving (Fei et al., 2024; Li et al., 2025b), human-scene reconstruction (Müller et al., 2025; Liu et al., 2025d), and 3D spatial reasoning (Fan et al., 2025).

**Stream 2: Diffusion-based 3D GFMs.**

Another line of work leverages diffusion models to reconstruct per-pixel 3D point maps, and in some cases, camera poses, through iterative denoising. DiffusionSfM (Zhao et al., 2025) parameterizes scene geometry and cameras as pixel-wise ray origins and endpoints in a global frame and directly infers 3D scene geometry and camera poses from multi-view images. Extending this idea to video, recent methods build on video diffusion models to tackle the more complex task of dynamic 4D reconstruction, where temporal consistency and object motion introduce additional challenges. Aether (Team et al., 2025) generates depth and ray maps via a diffusion backbone. GeometryCrafter (Xu et al., 2025) introduces a VAE-based architecture with dual encoder-decoders to improve pointmap quality from videos. Geo4D (Jiang et al., 2025b) unifies multiple geometric modalities—including points, rays, and depths—for temporally coherent reconstruction from video.

UniGeo (Sun et al., 2025) finetunes video diffusion model to predict geometry attributes such as surface normals and coordinates.

**End-to-End Sparse SfM.** There is also a parallel line of work on sparse 3D reconstruction in an end-to-end manner, including VGGSfM (Wang et al., 2024a) and FlowMap (Smith et al., 2024), which aim to replace classical SfM pipelines. While promising, these methods primarily focus on sparse reconstruction and are not the focus of our benchmark, which centers on dense 3D geometry prediction.

## A.2  MONOCULAR IMAGE/VIDEO DEPTH ESTIMATORS.

Before the emergence of 3D geometric foundation models (GFMs), numerous models were developed to estimate monocular depth from single images or videos. MiDaS (Ranftl et al., 2020) introduced affine-invariant supervision and training on mixed datasets for better zero-shot generalization. Depth Anything (Yang et al., 2024c) and its V2 (Yang et al., 2024d) extended this framework using transformer architectures and large-scale unlabeled data for semi-supervised learning. Pioneered by Marigold (Ke et al., 2024), recent works (Garcia et al., 2025; Fu et al., 2024; Gui et al., 2024; He et al., 2024; Pham et al., 2024) adapted pretrained diffusion models by finetuning only the U-Net on latent depth codes, converting depth maps to pseudo-RGB representations. To generalize to videos, temporal consistency was achieved through test-time optimization, memory modules, or stabilization networks (Zhang et al., 2021; Yasarla et al., 2023; Wang et al., 2023; Kopf et al., 2021; Luo et al., 2020b), and more recently by directly finetuning video diffusion models (Hu et al., 2024b; Shao et al., 2024; Yang et al., 2024c). While effective at producing dense depth, these models do not predict camera intrinsics or poses, and their affine-invariant outputs limit use in full 3D reconstruction.

Some methods explicitly address scale ambiguity and camera awareness. LeReS (Yin et al., 2021; 2022) incorporates 3D point cloud encoders to recover missing focal length and shift. UniDepth (Piccinelli et al., 2024) decouples camera parameter prediction from depth estimation via a pseudo-spherical representation and self-prompting. DepthPro (Bochkovskii et al., 2024) introduces a ViT-based model with a focal length prediction head, while MoGe (Wang et al., 2024b) uses affine-invariant point maps to reduce focal-distance ambiguity. Metric3D (Yin et al., 2023) and its V2 (Hu et al., 2024a) estimate metric depth using known camera parameters. Although these approaches improve monocular depth estimation, they can only process a single image and cannot be used to image collections or videos for 3D reconstruction directly.

Thus, we treat monocular depth estimators as valuable baselines for evaluating depth capabilities but do not consider them full GFMs in our benchmark.

## A.3  PRIOR BENCHMARKS IN 3D RECONSTRUCTION.

Classic 3D benchmarks such as DTU (Jensen et al., 2014), Tanks and Temples (Knapitsch et al., 2017), ETH3D (Schops et al., 2017a), KITTI (Geiger et al., 2012a;b), and ScanNet (Dai et al., 2017; Yeshwanth et al., 2023), have significantly advanced traditional pipelines for multi-view stereo (MVS) (Schönberger et al., 2016), structure-from-motion (SfM) (Schönberger & Frahm, 2016), and SLAM (Kerl et al., 2013). However, these benchmarks are typically task-specific and constrained to narrow domains: DTU focuses on object-centric tabletop scenes, KITTI on outdoor street driving, and ScanNet on indoor environments. Moreover, they are designed for systems that require intermediate inputs such as camera intrinsics, keypoint matches, or depth maps at inference time.

In contrast, 3D geometric foundation models (GFMs) operate in an end-to-end manner, predicting geometry directly from RGB inputs without relying on such precomputed representations. While existing benchmarks still offer valuable ground-truth data for evaluation, they are not sufficient to assess GFM performance across diverse capture conditions and tasks.

To address these limitations, our benchmark complements prior efforts by providing a unified evaluation framework that examines both the effectiveness and efficiency of modern GFMs. For effectiveness, it spans five core tasks: sparse-view and video-based depth estimation, extremely sparse and dense 3D reconstruction, relative pose estimation, and novel view synthesis. It also expands domain coverage to include dynamic real-world scenes, aerial drone footage, and challenging air–ground paired trajectories. For efficiency, we benchmark inference latency and peak memory usage under

1080 varying input scales. Together, these components form a comprehensive testbed for evaluating the
1081 generalization capabilities and deployment readiness of today's 3D geometric foundation models.

## B TRAINING CONFIGURATIONS ACROSS EVALUATED GFMS

While E3D-Bench is designed purely for inference-time evaluation, understanding the training
configurations and hyperparameters of the evaluated GFMs is essential for interpreting performance
differences. Many observed performance gaps likely stem not from architectural superiority alone,
but from variations in training data, loss functions, supervision signals, or fine-tuning strategies.
To facilitate deeper insights and guide future model development, we expand Tab. 1 with detailed
training configurations in Tab. S1.

Table S1: **Training configuration comparisons of evaluated end-to-end 3D geometric foundation models.**
Methods are grouped by input type.

| Method | Train Datasets | Architecture | Train Loss |
|---|---|---|---|
| **Pair of Images** | | | |
| DUSt3R | Habitat, MegaDepth, ARKitScenes, Static Scenes 3D, BlendedMVS, ScanNet++, CO3D-v2, Waymo | ViT-L–ViT-B | PM |
| MASt3R | Habitat, ARKitScenes, BlendedMVS, MegaDepth, Static Scenes 3D, ScanNet++, CO3D-v2, Waymo, Mapfree, WildRGB, Virtual KITTI, Unreal4K, TartanAir, internal | ViT-L–ViT-B | PM, M |
| LSM | ScanNet, ScanNet++ | ViT-L–ViT-B | PM, Ph, S |
| MonST3R | PointOdyssey, TartanAir, Spring, Waymo Perception | ViT-L–ViT-B | PM, TS, F |
| NoPoSplat | RealEstate10K; ACID; DL3DV | ViT-L–ViT-B | L2, LP |
| Align3R | SceneFlow, Virtual KITTI, TartanAir, Spring, PointOdyssey | ViT-L–ViT-B | PM |
| Splatt3R | ScanNet++ | ViT-L–ViT-B | L2, LP |
| Easi3R | — | ViT-L–ViT-B | — |
| **Image Sequences** | | | |
| Spann3R | Habitat, ScanNet, ScanNet++, ARKitScenes, BlendedMVS, CO3D-v2 | ViT-L–ViT-B | PM, Sc |
| CUT3R | ARKitScenes, BlendedMVS, CO3D-v2, MegaDepth, ScanNet++, ScanNet, Waymo, WildRGB-D, Map-free, TartanAir, UnrealStereo4K, Virtual KITTI 2, 3D Ken Burns, BEDLAM, COP3D, DL3DV, Dynamic Replica, EDEN, Hypersim, IRS, Matterport3D, MVImgNet, MVS-Synth, OmniObject3D, PointOdyssey, RealEstate10K, Smart-Portraits, Spring, Synscapes, UASOL, UrbanSyn, HOI4D | ViT-L–ViT-B | PM, P, R |
| Aether | Self-generated synthetic 4D data | CogVideoX-5B-I2V | L2, MS, D, PM |
| Geo4D | Spring, BEDLAM, PointOdyssey, TartanAir, Virtual KITTI | DynamiCrafter | PM, Di, C, TS |
| GeometryCrafter | 3D Ken Burns, Dynamic Replica, GTA-SfM, Hypersim, IRS, Matrix-City, MidAir, Spring, Structured3D, SYNTHIA, TartanAir, UrbanSyn, Virtual KITTI 2 | SVD | PM, N, MSD |
| **Multi-view Images** | | | |
| Fast3R | CO3D, ScanNet++, ARKitScenes, Habitat, BlendedMVS, MegaDepth | ViT-L–ViT-L | PM, D |
| VGGT | CO3D-v2, BlendedMVS, DL3DV, MegaDepth, Kubric, WildRGB, ScanNet, Hypersim, Mapillary, Habitat, Replica, MVS-Synth, PointOdyssey, Virtual KITTI, Aria Synthetic Environments, Aria Digital Twin, artist-created synthetic data | DINOv2–Heads | PM, D, T, C |
| Mapanything | BlendedMVS, Mapillary Planet-Scale Depth, ScanNet++ v2, Spring, TartanAirV2-WB, UnrealStereo4K, Aria Synthetic Environments, DL3DV-10K, Dynamic Replica, MegaDepth, MVS-Synth, ParallelDomain-4D, SAIL-VOS 3D | DINOv2–Heads | PM, RD, Q, t, RDp, LPM, MScl, N, GM, H |
| **Sparse-view Images** | | | |
| FLARE | MegaDepth, ARKitScenes, BlendedMVS, ScanNet++, CO3D-v2, Waymo, WildRGB-D, DL3DV | ViT-L–ViT-B | PM, P, L2, VGG, 3DGS-D |

**Loss legend:** PM = Pointmap regression; D = Depth; M = Matching; P = Pose; T = Tracking; C = Camera; R = RGB; L2 = MSE; LP = LPIPS; VGG = VGG perceptual; Ph = Photometric; S = Semantic; TS = Trajectory smoothness; F = Flow projection; Di = Disparity; N = Surface normals; MS = MS-SSIM; MSD = Multi-scale depth; 3DGS-D = 3D Gaussian Splatting rendered depth; RD = Ray direction; Q = Pose quaternions; $t$ = Translation; RDp = Ray depth; LPM = Local pointmap; MScl = Metric scale; Sc = Scale (regularization); GM = Gradient matching; H = Entropy.

## C BENCHMARK DETAILS

Tab.S2 summarizes the datasets and preprocessing procedures used in our benchmark. Detailed
evaluation protocols for sparse-view and video depth estimation are provided Sec. C.1 and Sec. C.2
respectively, and evaluation details of novel view synthesis in Sec. C.3. Implementation details,
including model input and resolution setup, are described in Sec. C.4.

Table S2: **Summary of Benchmark Tasks, Datasets, Number of Scenes, Domain, and Preprocessing**.

| Task | Dataset | # of Scenes | Domain | Dataset Preprocessing |
|---|---|---|---|---|
| Sparse-View Depth Estimation | DTU (Jensen et al., 2014) | 110 | Object-Centric | Following RobustMVD protocol |
| | ScanNet (Dai et al., 2017) | 200 | Indoor | |
| | KITTI (Geiger et al., 2012a) | 93 | Outdoor/Driving | |
| | ETH3D (Schops et al., 2017a) | 104 | Indoor/Outdoor | |
| | Tanks and Temples (Knapitsch et al., 2017) | 69 | Indoor/Outdoor | |
| Video Depth Estimation | Bonn (Palazzolo & Leutenegger, 2019) | 5 | Indoor | Frames 30–140; stride=2 |
| | TUM Dynamics (Sturm et al., 2012b) | 8 | Indoor | stride=30 |
| | KITTI (Geiger et al., 2012a) | 13 | Outdoor/Driving | First 110 frames; stride=2 |
| | PointOdyssey val (Zhao et al., 2024) | 15 | Synthetic (Indoor/Outdoor) | First 110 frames; stride=2 |
| | Syndrone (Rizzoli et al., 2023) | 8 | Aerial (Synthetic) | First 100 frames; stride=2 |
| | Sintel (Butler et al., 2012) | 14 | Synthetic (Indoor/Outdoor) | All frames used |
| Multi-View Relative Pose Estimation | CO3Dv2 (Ruiz et al., 2022) | 2511 | Object-Centric | Randomly selected 10 frames |
| | RealEstate10K (Zhou et al., 2017) | 1611 | Indoor/Outdoor | Randomly selected 10 frames |
| | ScanNet-eval (Dai et al., 2017) | 89 | Indoor | First 1200 frames; stride=30 |
| | Bonn (Palazzolo & Leutenegger, 2019) | 5 | Indoor/Outdoor | Frames 30–140; stride=2 |
| | TUM Dynamics (Sturm et al., 2012b) | 8 | Indoor | stride=30 |
| | KITTI Odometry (Geiger et al., 2012b) | 11 | Outdoor/Driving | First 110 frames; stride=2 |
| | Sintel (Butler et al., 2012) | 14 | Synthetic (Indoor/Outdoor) | All frames used |
| | ADT (Koppula et al., 2024) | 8 | Indoor | stride=10 |
| | ACID (Zhou et al., 2022) | 1500 | Aerial (Outdoor) | Randomly selected 10 frames |
| | Syndrone (Rizzoli et al., 2023) | 8 | Aerial (Synthetic) | First 100 frames; stride=2 |
| | ULTRRA (Joshi et al., 2024) | 4 | Aerial-Ground Pair | Following Vuong et al. (2025) |
| Multi-View 3D Reconstruction (Extremely Sparse) | DTU (Jensen et al., 2014) | 22 | Object-Centric | stride=16 |
| | 7-Scenes (Shotton et al., 2013) | 18 | Indoor | stride=200 |
| | NRGBD (Yang et al., 2023) | 9 | Indoor | stride=500 |
| | ScanNet (Dai et al., 2017) | 20 | Indoor | First 1200 frames; stride=300 |
| | TUM RGBD (Sturm et al., 2012a) | 11 | Indoor | First 1200 frames; stride=300 |
| Multi-View 3D Reconstruction (Dense) | DTU (Jensen et al., 2014) | 22 | Object-Centric | stride=5 |
| | 7-Scenes (Shotton et al., 2013) | 18 | Indoor | stride=30 |
| | NRGBD (Yang et al., 2023) | 9 | Indoor | stride=40 |
| | ScanNet (Dai et al., 2017) | 20 | Indoor | First 1200 frames; stride=30 |
| | TUM RGBD (Sturm et al., 2012a) | 11 | Indoor | First 1200 frames; stride=30 |
| Novel View Synthesis | DTU (Jensen et al., 2014) | 16 | Object-Centric | Following NoPoSplat protocol |
| | RealEstate10K (Zhou et al., 2017) | 34 | Indoor/Outdoor | |
| | ScanNet++ (Yeshwanth et al., 2023) | 50 | Indoor | |
| | ACID (Zhou et al., 2022) | 11 | Aerial (Outdoor) | |

## C.1 Sparse-View Depth Estimation

**Additional Evaluation Details:** Depth maps are extracted from the z-coordinate of predicted pointmaps after projection into the camera coordinate system. For models that output both camera- and world-coordinate pointmaps, we use the pointmaps in the world coordinate system to ensure consistency.

For GFMs requiring global alignment to handle multi-view inputs, we extract depth maps after alignment. Specifically, DUSt3R and MASt3R perform global alignment using complete scene graphs, while MonST3R adopts a sliding-window strategy. However, MonST3R's flow loss occasionally produces invalid values (e.g., NaN), particularly on static datasets. In such cases, we disable the flow loss module to maintain evaluation integrity.

To isolate geometry quality from camera estimation errors, we use ground-truth camera intrinsics and poses for all models. To fairly compare models with different output scales, we report results under both normalized and metric settings: 1) For normalized-scale models, we apply per-view median alignment, using the ratio between predicted and ground-truth median depths. 2) For metric-scale models, we report two variants: (i) raw predictions, and (ii) the same median-aligned depths used for normalized models, enabling consistent comparisons.

To control for variations due to input view selection, we follow the quasi-optimal source view strategy proposed in RobustMVD (Schroppel et al., 2022). If global alignment fails to converge with quasi-optimal views, we fall back to using the nearest neighboring views to ensure evaluation continuity.

## C.2 Video Depth Estimation

**Additional Evaluation Details:** We evaluate both normalized- and metric-scale models. For normalized-scale models, we apply a scale-and-shift alignment to ground truth before computing metrics. For metric-scale models, we report two variants: (i) using raw predictions, and (ii) applying the same scale-and-shift alignment for fair comparison. To consistently evaluate temporal coherence, we follow MonST3R and compute a single global scale and shift per video sequence—rather than per frame—before metric evaluation. This sequence-level alignment ensures that temporal consistency is not biased by framewise normalization. It should be noted that for Geo4D, with the input sequences containing fewer than 16 frames, we pad them to 16 frames by repeating the last frame of the input video. For Aether, we use its default one-shot output of 41 frames for evaluation. The same settings are applied in all subsequent experiments.

## C.3 Novel View Synthesis

**Additional Evaluation Details:** For DUSt3R variants equipped with appearance modeling (i.e., LSM, Splatt3R, NoPoSplat, and FLARE), training is typically performed on narrow domain-specific datasets, for example, Splatt3R is trained only on ScanNet++, and NoPoSplat on ACID or RealEstate10K. Unlike methods such as PixelSplat (Charatan et al., 2023) and MVSplat (Chen et al., 2024a), these models are all pose-free: given a pair of images, they directly predict per-pixel 3D Gaussians without requiring ground-truth camera intrinsics or extrinsics. The resulting scenes are scale-invariant, meaning ground-truth novel-view poses cannot be directly applied at inference time. To address this, NoPoSplat and FLARE optimize the test-time novel pose via photometric losses (PSNR + LPIPS), while LSM and Splatt3R rescale the ground-truth pose using the relative scale between predicted and ground-truth pointmaps.

Since not all of our selected datasets provide ground-truth depth, we apply test-time pose optimization for all methods to ensure a fair and consistent comparison. Splatt3R is not included in this comparison because its evaluation code was not publicly released at the time of writing. All evaluations are conducted at a resolution of $256 \times 256$, and the results are summarized in Tab. 6. For NoPoSplat, which provides two separately trained models on ACID and RealEstate10K, we follow the official instructions and use the corresponding model for each respective test set and use the model trained on RealEstate10K for all other datasets by default to maintain consistency across evaluations.

### C.4 IMPLEMENTATION DETAILS

GFMs vary in their input resolution requirements—for example, VGGT requires both height and width to be divisible by 14; DUSt3R and MASt3R typically use $512\times384$; Spann3R uses $224\times224$. To preserve each model's intended behavior, we adopt its native input resolution. However, for fair comparison across tasks, we standardize output resolution: we use the original resolution for sparse depth estimation and $512\times384$ for all other tasks. For diffusion-based methods, which require a fixed number of input frames (e.g., Geo4D uses 16 frames), we pad shorter sequences by repeating the last frame to meet the input requirement. We omit 3D reconstruction evaluation for all diffusion models, and pose estimation for GeometryCrafter, as these tasks are not natively supported by those architectures.

## D MORE QUANTITATIVE AND QUALITATIVE RESULTS

### D.1 VIDEO DEPTH ESTIMATION

In Tab. 3 in the main draft, we observe that current GFMs perform competitively, and in some cases surpass, specialized video depth models such as DepthCrafter and Marigold. In Fig. S1, we visualize results from four top-performing GFMs (VGGT, Align3R, Geo4D, and Aether), alongside the baseline VideoDepthAnything.

We observe that: 1) VGGT captures fine-grained geometric relationships, such as the relative depth between the chair and arm in TUM Dynamics, relative depth between hand and apple in Sintel, or the presence of a streetlamp in Syndrone. Although its predictions may appear slightly blurred at object boundaries, its depth structure is semantically accurate. 2) Align3R produces sharp object silhouettes (e.g., monitor edges, chair legs) but exhibits less smoothness in distant regions, particularly in outdoor settings like Syndrone. 3) Geo4D and Aether (diffusion-based GFMs) yield crisp object contours but occasionally miss finer details, likely due to their iterative denoising formulation.

These qualitative results highlight that GFMs, despite not being explicitly designed for video depth, can rival or outperform task-specific approaches across a range of scenes.

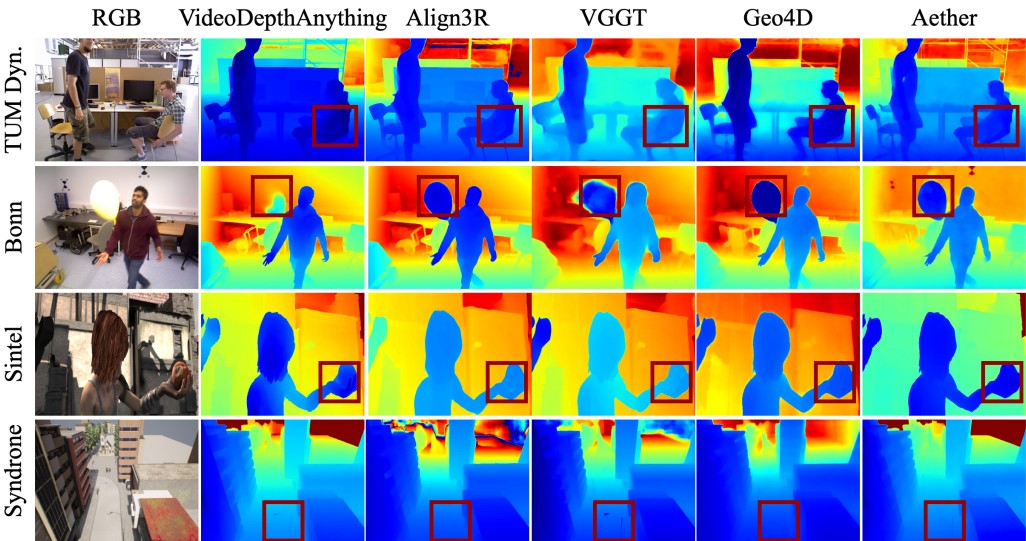

Figure S1: **Qualitative Comparison of Video Depth Estimation**.

### D.2 MULTI-VIEW RELATIVE POSE ESTIMATION

**Per-Dataset Breakdown:** We evaluate pose estimation across 11 datasets and group them into six representative settings: *In-Distribution* (CO3Dv2), *Long Sequence* (ScanNet, ADT, and TUM-Dynamics), *Street Driving* (KITTI Odometry), *Indoor-Outdoor* (Bonn, Sintel, and RealEstate10K),

*Drone* (ACID, Syndrone), and *Air-Ground* (ULTRRA Challenge). In Tab.4 of the main paper, we report scene-level averages for each setting to mitigate the effect of varying metric scales across datasets. For finer-grained comparison, we include a per-dataset breakdown for the *Long Sequence*, *Indoor-Outdoor*, and *Drone* categories in Tab. S3.

Table S3: **Per-Dataset Breakdown of Multi-view Relative Pose Estimation across Diverse Scenarios.** We report ATE ($\downarrow$), RPE translation ($\downarrow$), and RPE rotation ($\downarrow$) for each dataset.

(a) **Long Sequence Scenarios**

| Method | ScanNet | | | ADT | | | TUM-Dynamics | | |
|---|---|---|---|---|---|---|---|---|---|
| | ATE $\downarrow$ | RPE$_{trans}$ $\downarrow$ | RPE$_{rot}$ $\downarrow$ | ATE $\downarrow$ | RPE$_{trans}$ $\downarrow$ | RPE$_{rot}$ $\downarrow$ | ATE $\downarrow$ | RPE$_{trans}$ $\downarrow$ | RPE$_{rot}$ $\downarrow$ |
| DUSt3R/LSM | 0.098 | 0.071 | 2.072 | 0.653 | 0.454 | 5.651 | 0.077 | 0.089 | 2.709 |
| MASt3R | 0.086 | 0.067 | 2.112 | 0.663 | 0.439 | 5.628 | 0.098 | 0.102 | 8.790 |
| Spann3R | 0.295 | 0.144 | 3.764 | 0.520 | 0.495 | 6.147 | 0.064 | 0.054 | 1.564 |
| CUT3R | 0.151 | 0.112 | 3.917 | 0.675 | 0.436 | 5.635 | 0.065 | 0.069 | 9.474 |
| VGGT | 0.070 | 0.060 | 1.249 | 0.694 | 0.439 | 5.673 | 0.015 | 0.018 | 0.586 |
| Fast3R | 0.513 | 0.405 | 26.286 | 0.737 | 0.508 | 9.720 | 0.106 | 0.116 | 9.424 |
| Mapanything | 0.151 | 0.115 | 2.864 | 0.657 | 0.427 | 5.601 | 0.053 | 0.062 | 3.062 |
| MonST3R | 0.463 | 0.258 | 12.947 | 0.535 | 0.739 | 10.228 | 0.192 | 0.146 | 13.957 |
| Align3R | 0.431 | 0.206 | 9.487 | 0.674 | 0.440 | 6.256 | 0.103 | 0.086 | 11.965 |
| Easi3R | 0.135 | 0.074 | 2.569 | 0.687 | 0.447 | 5.830 | 0.100 | 0.086 | 3.282 |
| Geo4D | 0.428 | 0.156 | 10.486 | 0.793 | 0.423 | 9.408 | 0.176 | 0.139 | 12.593 |
| Aether | 0.681 | 0.283 | 15.555 | 0.704 | 0.328 | 8.717 | 0.174 | 0.111 | 12.535 |

(b) **Indoor-Outdoor Scene**

| Method | Bonn | | | Sintel | | | RealEstate10k | | |
|---|---|---|---|---|---|---|---|---|---|
| | ATE $\downarrow$ | RPE$_{trans}$ $\downarrow$ | RPE$_{rot}$ $\downarrow$ | ATE $\downarrow$ | RPE$_{trans}$ $\downarrow$ | RPE$_{rot}$ $\downarrow$ | ATE $\downarrow$ | RPE$_{trans}$ $\downarrow$ | RPE$_{rot}$ $\downarrow$ |
| DUSt3R/LSM | 0.026 | 0.022 | 1.259 | 0.355 | 0.202 | 13.740 | 0.075 | 0.562 | 1.553 |
| MASt3R | 0.022 | 0.022 | 1.245 | 0.340 | 0.291 | 5.977 | 0.056 | 0.563 | 1.265 |
| Spann3R | 0.041 | 0.015 | 1.721 | 0.329 | 0.114 | 4.114 | 0.081 | 0.102 | 1.272 |
| CUT3R | 0.033 | 0.015 | 1.150 | 0.209 | 0.071 | 0.634 | 0.031 | 0.039 | 0.497 |
| VGGT | 0.013 | 0.016 | 1.118 | 0.172 | 0.062 | 0.471 | 0.061 | 0.111 | 0.579 |
| Fast3R | 0.034 | 0.035 | 1.594 | 0.267 | 0.209 | 14.342 | 0.110 | 0.170 | 1.911 |
| Mapanything | 0.025 | 0.018 | 1.199 | 0.210 | 0.092 | 2.773 | 0.058 | 0.108 | 0.575 |
| MonST3R | 0.023 | 0.013 | 1.105 | 0.107 | 0.039 | 0.664 | 0.098 | 0.154 | 0.831 |
| Align3R | 0.022 | 0.013 | 1.115 | 0.542 | 0.150 | 0.602 | 0.072 | 0.091 | 1.088 |
| Easi3R | 0.019 | 0.016 | 1.308 | 0.370 | 0.212 | 13.726 | 0.073 | 0.093 | 1.254 |
| Geo4D | 0.026 | 0.012 | 1.113 | 0.179 | 0.064 | 0.515 | 0.578 | 0.477 | 3.816 |
| Aether | 0.019 | 0.012 | 0.863 | 0.158 | 0.046 | 0.632 | 0.195 | 0.123 | 1.621 |

(c) **Drone Scene**

| Method | ACID | | | Syndrone | | |
|---|---|---|---|---|---|---|
| | ATE $\downarrow$ | RPE$_{trans}$ $\downarrow$ | RPE$_{rot}$ $\downarrow$ | ATE $\downarrow$ | RPE$_{trans}$ $\downarrow$ | RPE$_{rot}$ $\downarrow$ |
| DUSt3R/LSM | 0.124 | 0.368 | 2.849 | 0.535 | 2.565 | 0.300 |
| MASt3R | 0.129 | 0.364 | 2.614 | 0.324 | 2.542 | 0.214 |
| Spann3R | 0.108 | 0.136 | 1.484 | 1.817 | 2.706 | 1.398 |
| CUT3R | 0.062 | 0.077 | 0.917 | 1.684 | 2.565 | 0.311 |
| VGGT | 0.280 | 0.450 | 0.806 | 0.206 | 2.532 | 0.109 |
| Fast3R | 0.431 | 0.505 | 1.984 | 1.537 | 3.119 | 1.026 |
| Mapanything | 0.276 | 0.582 | 2.826 | 0.549 | 2.565 | 0.291 |
| MonST3R | 0.321 | 0.493 | 1.519 | 2.966 | 2.563 | 0.594 |
| Align3R | 0.144 | 0.167 | 0.981 | 1.208 | 2.528 | 0.133 |
| Easi3R | 0.109 | 0.125 | 1.740 | 2.012 | 2.560 | 0.256 |
| Geo4D | 0.376 | 0.317 | 1.400 | 1.795 | 2.550 | 0.434 |
| Aether | 0.146 | 0.088 | 0.796 | 1.415 | 1.759 | 0.727 |

**Blue** : Long Sequence  **Orange** : Indoor-Outdoor Scene  **Cyan** : Drone

We observe several trends in the results: 1) VGGT demonstrates the best generalization, dominating the long sequence scenarios and drone footage. 2) In addition to DUSt3R and MASt3R, methods designed for online registration, such as Spann3R and CUT3R, also perform well on long video sequences. 3) Among diffusion-based models, Aether consistently ranks among the top performers, while Geo4D shows strong results primarily in indoor-outdoor scenes.

**Detailed Comparison on the ULTRRA Challenge:** Our pose estimation metrics are computed using Sim(3) Umeyama alignment between predicted and ground-truth trajectories. However, in the ULTRRA dataset, the aerial and ground trajectories are reconstructed in separate coordinate frames. Despite being calibrated with RTK-corrected GPS, the absence of a shared reference system makes a single global alignment infeasible, rendering ATE inapplicable. As a result, we omit ATE and only

report RPE-trans and RPE-rot in Tab.4 of the main paper. Notably, both metrics are significantly worse than in all other settings, making it difficult to discern which GFMs perform well in this challenging domain.

To supplement these results, in Tab. S4, we report RTA@$\tau$ and RRA@$\tau$: the percentage of camera pairs with relative translation or rotation error below threshold $\tau$. These metrics better reflect performance under the unique challenges of this setting. Diffusion-based methods (e.g., Geo4D, Aether) are excluded from this evaluation, as they require fixed-length input sequences (e.g., 16 and 41 frames respectively), making them incompatible with the pairwise-view setup in ULTRRA.

Table S4: **Camera Rotation and Translation Accuracy on ULTRRA challenge** (excluding fine-tuned models from top-3 ranking).

| Method | Camera Rotation Accuracy | | | Camera Translation Accuracy | | |
|---|---|---|---|---|---|---|
| | RRA@5 ↑ | RRA@10 ↑ | RRA@15 ↑ | RTA@5 ↑ | RTA@10 ↑ | RTA@15 ↑ |
| DUSt3R (ft AerialMegaDepth) | 55.96 | 71.25 | 76.15 | 46.48 | 68.20 | 72.78 |
| MASt3R (ft AerialMegaDepth) | 49.54 | 66.36 | 72.48 | 42.51 | 63.30 | 69.11 |
| DUSt3R | 5.50 | 8.56 | 9.79 | 2.45 | 6.12 | 8.26 |
| MASt3R | 3.06 | 3.98 | 5.50 | 1.83 | 3.67 | 4.89 |
| VGGT | 6.73 | 10.09 | 13.15 | 1.53 | 4.89 | 6.42 |
| Fast3R | 5.50 | 12.84 | 22.02 | 2.45 | 7.34 | 11.01 |
| Mapanything | 12.23 | 16.51 | 19.27 | 1.22 | 4.28 | 5.81 |
| Spann3R | 9.79 | 26.30 | 38.53 | 4.28 | 8.87 | 15.90 |
| MonST3R | 1.83 | 2.75 | 3.67 | 0.31 | 1.83 | 2.75 |
| CUT3R | 3.67 | 16.51 | 22.02 | 0.31 | 2.45 | 5.20 |
| Align3R | 4.11 | 9.89 | 12.08 | 1.71 | 3.85 | 6.27 |
| Easi3R | 4.70 | 10.26 | 13.68 | 1.28 | 5.13 | 8.97 |

From these results, we observe: 1) Without fine-tuning on datasets like AerialMegaDepth, Spann3R achieves the best accuracy, followed closely by Fast3R. 2) Fine-tuning clearly provides substantial performance gains for this challenging out-of-distribution scenario, highlighting the importance of domain-specific adaptation to improve the domain generalization of GFMs, as mentioned in our findings.

To better understand common failure modes in ULTRRA, we also visualize the reconstruction results of Fast3R, MapAnything, and AerialMegaDepth in Figure S2. These examples reveal that GFMs frequently misplace ground-view reconstruction due to the extreme domain shift.

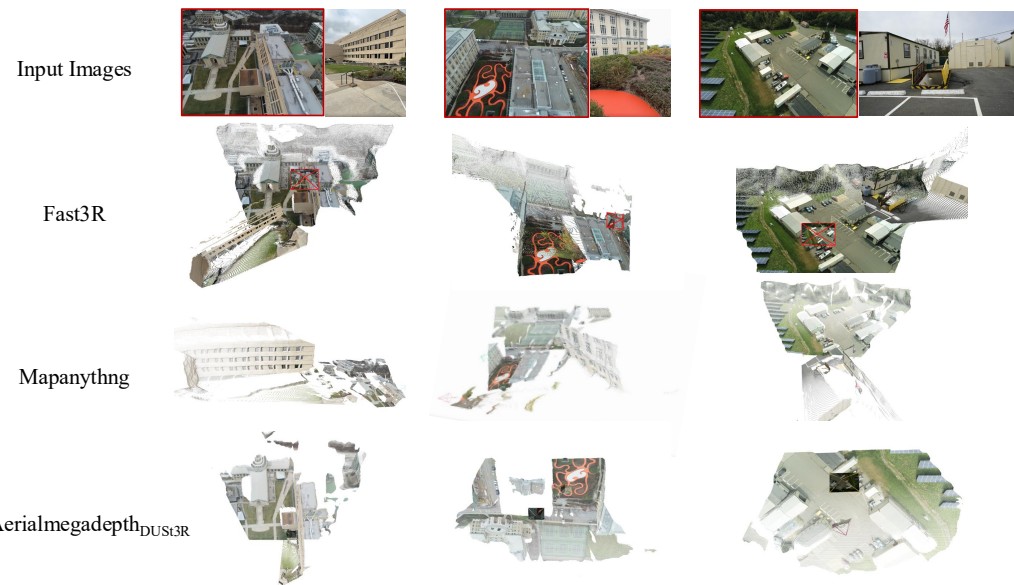

Figure S2: **Qualitative Comparison of ULTRRA Challenge**. We omit Spann3R from visualization due to its lower input resolution (224×224). Red boxes indicate the aerial-view inputs.

## D.3 NOVEL VIEW SYNTHESIS

Novel view synthesis results on DTU, ScanNet++, RealEstate10K, and ACID are shown in Fig. S3. We observe the following: 1) LSM often produces visible holes in the synthesized views (e.g., the wall in RealEstate10K), and struggles with fine-grained structures, (e.g., thin floor lamps in ScanNet++), possibly due to the erroneous pointmap prediction. 2) When using test-time pose optimization, the optimized pose may still fail to perfectly align with ground-truth, especially on datasets like DTU. This misalignment is evident across all methods and helps explain their relatively lower performance on DTU in Tab. 6.

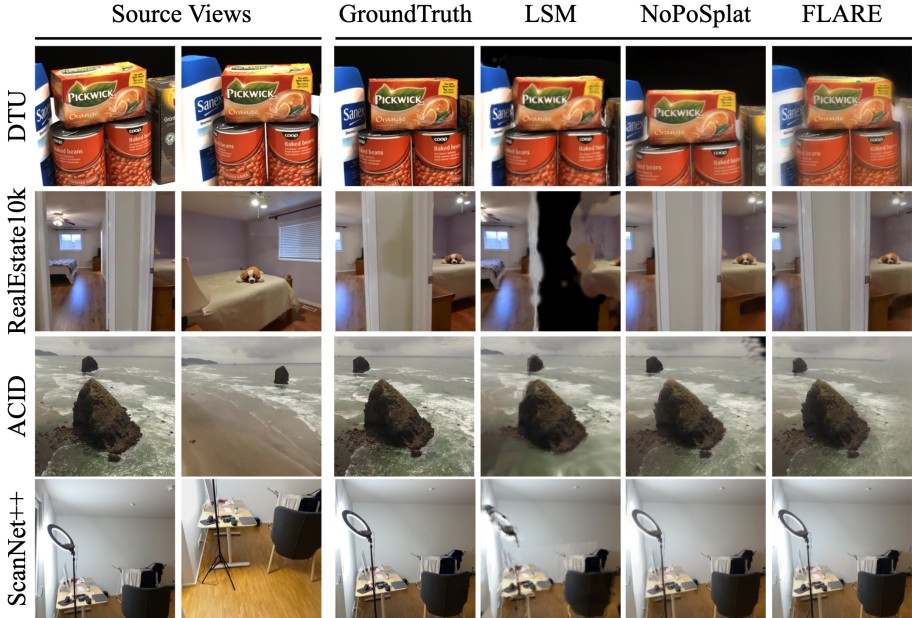

Figure S3: **Qualitative Comparison of Novel View Synthesis** given two source views. Rendering resolution is 256×256.

## D.4 EXTREMELY SPARSE-VIEW 3D RECONSTRUCTION

As noted in Tab. 5 of the main paper, VGGT, CUT3R, FLARE, DUSt3R, and MASt3R perform well in extremely sparse-view 3D reconstruction, while models such as MonST3R, Align3R, and Easi3R show noticeably lower performance (with MonST3R included as a representative example). To illustrate these differences, we visualize reconstructions from VGGT, CUT3R, DUSt3R, and MonST3R in Fig. S4. We observe the following: (1) For image sets with minimal or no overlap, VGGT consistently reconstructs clean and coherent 3D structures, whereas MonST3R often produces outliers. (2) DUSt3R performs especially well on the DTU dataset, which aligns with its strong quantitative results reported in Tab.5 in main draft.

## E BROADER IMPACT

End-to-end 3D GFMs have transformative applications in industry and society. (i) Advantages of integrating data-driven 3D perception into existing autonomous systems strengthen safety and robust decision-making, replacing fragmented pipelines with seamless, real-time frameworks for perception, planning, and control. (ii) New capabilities provided by real-time 3D understanding and generation systems in immersive AR/VR, robotics, and digital twins; (iii) Benefits of scalable 3D modeling in the scientific imaging domain, including computational imaging (CT), cryo-electron microscopy (cryo-EM), and magnetic resonance imaging (MRI), which require high precision in 3D reconstruction and pose estimation.

DUSt3R  MonST3R  CUT3R  VGGT

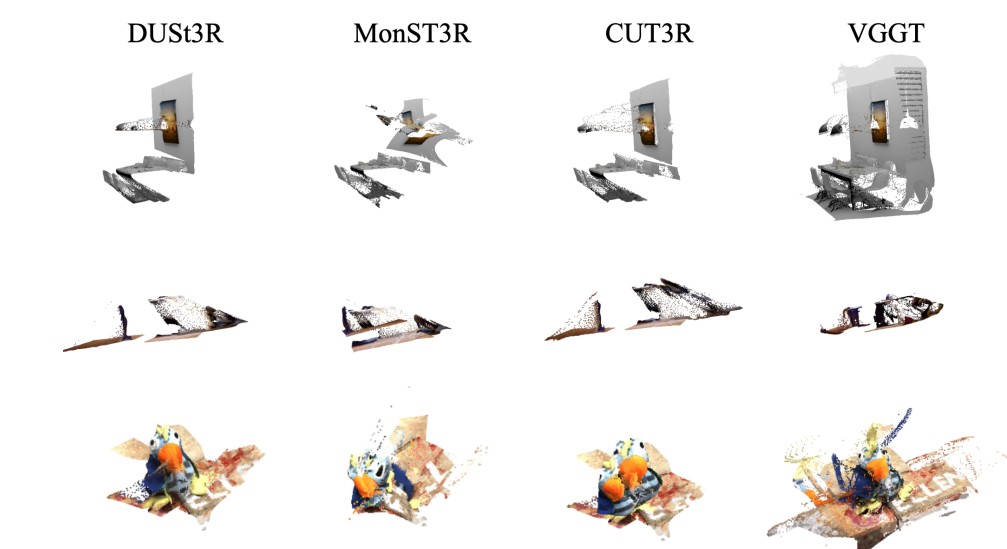

Figure S4: **Qualitative Comparison of Extremely Sparse-View 3D Reconstruction**. From top to bottom are scenes from NRGBD, TUM RGBD, and DTU.

## F    FUTURE DIRECTIONS

Geometric foundation models (GFMs) represent a promising foundation for a wide range of emerging research directions beyond traditional 3D reconstruction tasks. One important future trajectory is their use as a **data generator**. For example, recent works like Khazatsky et al. (2024); Li et al. (2025c) leverage GFMs' ability to calibrate cameras or predict consistent 3D geometry from arbitrary data sources (e.g., Internet videos, or egocentric streams) to produce high-quality 3D supervision for downstream tasks. Such synthetic 3D supervision could reduce reliance on expensive ground-truth annotations, especially for tasks where dense 3D labels are scarce or difficult to obtain.

Another compelling direction lies in using GFMs as **spatial priors or encoders** within multimodal foundation models. For instance, recent work like (Fan et al., 2025; Wu et al., 2025; Huang et al., 2025; Cao et al.) demonstrates how geometric representations can enhance visual-language models (VLMs) by enabling spatial reasoning, 3D-aware grounding, and multimodal alignment across views. Incorporating geometric understanding into large multimodal systems could improve their ability to reason about spatial relationships, perform grounded generation, and interpret complex physical scenes from limited visual input.

Looking ahead, we envision GFMs playing a central role in future embodied intelligence systems, serving as a unified backbone for spatial understanding, environment simulation, and interaction planning in real-world 3D environments.

