# OpenReview forum: "E3D-Bench: A Benchmark for End-to-End 3D Geometric Foundation Models"
_ICLR.cc/2026/Conference — ICLR 2026 Conference Desk Rejected Submission_

### Official Review · Reviewer_HUyc · 2025-10-29

**Soundness:** 3
**Presentation:** 3
**Contribution:** 3
**Rating:** 6
**Confidence:** 4

**Summary:**

This paper proposes a benchmark for recent popular geometry foundation models, and evaluates the performance on various tasks with consistent metrics. The evaluation is very comprehensive, including indicators such as depth estimation accuracy, pose accuracy, and reconstruction accuracy, etc. After evaluating multiple methods, the authors also provide several insights to inspire further research.

**Strengths:**

This paper provides a thorough evaluation on multiple geometry foundation models on a series of 3D related tasks, and further summarizes several key findings to help future researches.
1.The included GFMs are rich, including pair-based, multi-view-based, image-sequence-based, and even diffusion models.
2.The evaluation is comprehensive, including metrics such as depth estimation accuracy, pose accuracy, and reconstruction accuracy.
3.The findings are insightful for future researches on GFMs.

**Weaknesses:**

1. Training cost and deployment cost are metrics of concern for both researchers and industry. Quantitatively evaluating the relationship between performance and these costs is a potential direction for improvement in the E3D Benchmark. Training cost may include factors such as dataset collection and storage cost, GPU hours required for training, and memory consumption. Deployment cost, on the other hand, may include factors such as FLOPs and memory usage per inference.
2. In the network architecture design section, a more in-depth analysis can be conducted for feed-forward architectures, particularly regarding how different network architectures handle multi-view images. For example, this includes the GA module in DUSt3R/MASt3R, the memory mechanisms in spann3r and cut3r, as well as the direct inference strategies adopted by Fast3R and VGGT.

**Questions:**

No further questions for the authors. See weakness above.

---

> ### Author Response · Authors · 2025-11-22
> **Rebuttal by Authors**
>
> We sincerely thank the reviewer for endorsing E3D-Bench as a **"thorough"** and **“comprehensive”** evaluation that covers a **"rich"** set of GFMs across diverse paradigms, and recognition that our findings are **"insightful for future research"**. We value your constructive suggestions regarding cost analysis and architectural trade-offs, which we address below.
>
> ___
>
> ### **Q1: Quantifying training & deployment cost vs performance**
>
> We agree that quantifying the cost-performance trade-off is essential for real-world adoption.
>
> - **Deployment Cost:** E3D-Bench aims to help the multimodal learning and computer vision community to find the proper model for their downstreaming application with direct inference. Therefore we prioritize the **"off-the-shelf" inference** capability that most practitioners rely on. It
>
>    - We heavily emphasize practical deployment costs, specifically **Inference Latency** and **Peak GPU Memory vs. #views on A100** (Figure 2, Sec. 3.6). These metrics directly reflect the bottlenecks (memory bandwidth and OOM risks) encountered in production.
>    - We will add **# of Parameters** (model size) for all methods in Table 1 to provide a proxy for arithmetic complexity.
>
> - **Training Cost:** While we agree training efficiency is a key metric, a rigorous quantitative comparison is hindered by the following two factors. So we instead summarize available training profiles in Table S1 in the Appendix and leave a full training-cost analysis to future work.
>
>    - **Computational Feasibility:** A rigorous cost–performance study would require retraining all GFMs under a unified recipe, which is computationally prohibitive. For instance, VGGT alone needs 64×A100 for >9 days.
>    - **Reproducibility Barriers:** Several methods have not released complete training codes, preprocessing pipelines, or hyperparameter details.
>
> We will add these discussions into our revision.
>
> ---
>
> ### **Q2: Deeper analysis of multi-view handling**
>
> We appreciate the suggestion to deepen the architectural analysis. Based on our empirical results, we have identified consistent behavioral patterns across the three dominant multi-view handling strategies, which we will formalize in Sec 4 of the revision:
>
> ### • Global Alignment (GA) (e.g., DUSt3R, MASt3R)
>
> - **Method:** These models function by first predicting geometry for pairs of images and then employing a Global Alignment module. This module constructs a scene graph to optimization-basedly align pairwise pointmaps into a unified coordinate system.
>
> - **Performance Pattern:** They achieve high robustness on dense or well-overlapping views (e.g., DTU reconstruction, Tab. 5) due to the graph optimization. However, this optimization causes latency and memory usage to scale poorly (\$O(N^2)\$ or steep \$O(N)\$) with view count (Fig. 2).
>   They also degrade under extremely sparse or OOD settings where pairwise matching fails, destabilizing the global graph (Tabs 2–4).
>
> ### • Memory / Online Registration (e.g., Spann3R, CUT3R)
>
> - **Method:** These models utilize recurrent architectures or spatial memory modules to process sequences in a streaming fashion. They maintain a latent history state to propagate geometric information from previous views to the current frame, enabling online registration without full global optimization.
>
> - **Performance Pattern:** Designed for long sequences, they offer linear scaling and lower peak memory (Fig. 2), excelling at video depth and relative pose (Tabs 3–4).
>   However, due to the sequential nature of the memory, they often exhibit drift or weaker global consistency when view overlap is limited or the sequence is unordered, leading to less accurate sparse reconstruction (Tab. 5).
>
>
>
> ### • Direct Feed-Forward Multi-View (e.g., Fast3R, VGGT)
>
> - **Method:** These models employ efficient attention mechanisms (e.g., cross-view attention or voxel aggregation) to aggregate features from all input views simultaneously. They predict global geometry in a single network pass (one-shot) without iterative alignment or sequential recurrence.
>
> - **Performance Pattern:** They are the fastest and most scalable in terms of runtime (Fig. 2).
>   Their global context awareness makes them strong in sparse-view depth and cross-domain generalization (Tabs 2–3), with VGGT leading many tasks.
>   However, without the explicit constraints of GA optimization, their performance can drop on dense reconstruction tasks where pixel-perfect global consistency is required (Tab. 5).

---

### Official Review · Reviewer_oo1U · 2025-11-02

**Soundness:** 3
**Presentation:** 2
**Contribution:** 3
**Rating:** 6
**Confidence:** 4

**Summary:**

The paper introduces a thorough benchmark for 3D Geometric Foundation Models (GFMs), evaluating their performance across five different  tasks: sparse-view depth estimation, video depth estimation, 3D reconstruction, multi-view pose estimation, and novel view synthesis. It incorporates both conventional datasets and more challenging out-of-distribution scenarios, together with standardized evaluation protocols and metrics to ensure fair and reproducible comparisons. By analyzing 17 leading GFMs, the authors highlight each model’s strengths and limitations across diverse tasks and domains, offering valuable insights to guide future advancements in the field.

**Strengths:**

Following the seminal DUSt3R model, which introduced dense 3D representation prediction in a single feed-forward pass, numerous geometric foundation models (GFMs) have emerged, each proposing enhancements or variations of the original approach. A comprehensive and fair comparison of these methods across shared tasks, benchmark datasets, evaluation protocols, and metrics represents a highly valuable contribution to the research community.


Evaluating the models in the context of novel view synthesis (NVS) is a valuable addition especially considering cross-domain scenarios. It is somewhat disappointing that the authors had to limit this experiment to generating a third view given two input views.  It would be interesting to additionally explore how accurate vanilla Gaussian Splatting could be built using the predicted poses and the 3D point-clouds obtained by different GFM models. To evaluate such scenarios, for instance, target images could be registered (if needed) in the new reference coordinate frame. Then starting from the registered pose, the views rendered could be compared directly to the ground-truth target images. This would offer an interesting alternative assessment of the models' capabilities in NVS.

**Weaknesses:**

The paper includes a broad set of leading GFMs in its analysis, which is highly appreciated. However, it is somewhat unfortunate that MUSt3R (CVPR’25,ArXiv:2503.01661) and MV-DUSt3R+ (CVPR’25,ArXiv:2412.06974), two multi-view extensions of DUSt3R, were not considered in the study. Both have publicly released their code and demonstrated notable improvements over DUSt3R. Additionally, MV-DUSt3R+ introduces support for novel view synthesis (NVS) through lightweight prediction heads that regress 3D Gaussian attributes.

The paper is extremely dense, presenting results from a large number of experiment, which is on the one hand much appreciated. On the other hand, despite the authors’ efforts to summarize key insights, the overall presentation remains difficult to follow and, at times, somewhat monotonous. To enhance readability and better communicate the main findings, it would be beneficial to move Section 3 to the supplementary material as-is and instead expanding Section 4. This expanded section could highlight for each key observation the findings with simplified tables (by retaining only the top-performing methods globally or by showing results averaged across datasets). Additionally, it would be helpful if each original table included an average rank for each method across all metrics, making it easier to grasp overall performance at a glance. Illustrative plots like the ones in Figure 1 are particularly effective and could be used more extensively to support the narrative.

**Questions:**

In the context of depth estimation, it would have been insightful to include results derived from the camera coordinate pointmap when available. It’s also important to note that for pairwise metric models such as MAST3R, global alignment does not guarantee scale preservation, which may explain the poor performance observed when evaluated without normalization. Although these models are referred to as metric due to their training with metric losses, the scale precision remains approximate (see for example Table 4 in MUSt3R paper). Consequently, the use of ground-truth intrinsics can influence depth predictions, as evidenced by the significantly improved results after normalization. For these reasons, I believe the inclusion of the bottom rows showing raw results may be misleading, and I would recommend removing them.

Lines 187-189: To more accurately assess how well metric scale is preserved by these methods, it would be preferable to derive depth directly from the camera coordinate pointmap. For models like MASt3R, this could be achieved by averaging the pointmaps of all image pairs in which the target image serves as the reference frame.

In Table 5, the results for MASt3R and DUSt3R appear quite similar, whereas the original MASt3R paper reports a more pronounced performance gap (see DTU results in Table 4). This discrepancy suggests that only coarse global alignment based on pointmaps was applied, without the more precise alignment using feature matching. For methods allowing matching maybe it would be interesting to also add results for refined reconstructions.

It would be interesting to also add a column to table 1 with the size of the model (number of parameters).

To improve readability and to better highlight the differences between methods, It would be helpful to  increase the size of the plots in Figure 2.

---

> ### Author Response · Authors · 2025-11-22
> **Rebuttal by Authors**
>
> We sincerely thank the reviewer for recognizing E3D-Bench as a **"highly valuable contribution to the research community"** by providing **“comprehensive and fair comparison”** of “**emerged GFMs**” “**across shared tasks, benchmark datasets, evaluation protocols, and metrics**”. We are grateful for your constructive feedback on presentation and methodology, which we have addressed below.
>
> ---
>
> ### **Q1: Add GFM + 3DGS in NVS task to verify the pose and point cloud accuracy**
>
> - Our current NVS comparison focuses on GFMs that **directly predict Gaussian attributes**, so we can evaluate their **feed-forward appearance modeling** without any per-scene optimization.
>
> - We agree that using GFM outputs (poses + point clouds) to initialize a vanilla 3DGS pipeline is an interesting complementary view, and several recent works follow this direction. However, raw GFM poses and point clouds are often **imperfect** (e.g., Fast3R reports non-trivial pose errors in Table 6), and plugging them directly into 3DGS typically yields millions of points, blurry views (pose errors) and multiple overlapping layers (noisy geometry). In practice, 3DGS pipelines initialized from GFMs usually **jointly optimize poses and all Gaussian parameters**, which makes it **difficult to attribute the final NVS quality back to the original GFM predictions**.
>
> - To better isolate geometry and pose, we add an experiment where we initialize 3DGS from GFMs and **freeze the 3D positions and camera poses**, optimizing only appearance parameters for a small number of steps (following the recipe of InstantSplat [1]). Due to the limited rebuttal window, we use three representative GFMs as initialization and report PSNR / SSIM / LPIPS / ATE on Tanks & Temples 12 views as an alternative NVS assessment.
>
> - We could find that **VGGT+InstantSplat achieves the best rendering quality** across all perceptual metrics (PSNR 26.95 / SSIM 0.8679), outperforming MASt3R and MapAnything. Interestingly, this occurs even though **MASt3R achieves the most accurate camera poses** (lowest ATE of 0.0124 vs. ~0.021 for others). This decoupling suggests that while MASt3R excels at global registration, **VGGT yields a higher-fidelity point cloud structure** that provides a superior geometric substrate for appearance modeling, validation of our finding that VGGT is a robust generalist.
>
>
> | Method                  | PSNR  | SSIM   | LPIPS  | ATE    |
> |------------------------|-------|--------|--------|--------|
> | MASt3R+InstantSplat     | 26.13 | 0.8363 | 0.1764 | 0.0124 |
> | VGGT+InstantSplat       | 26.95 | 0.8679 | 0.1494 | 0.0210 |
> | MapAnything+InstantSplat | 25.47 | 0.8080 | 0.1912 | 0.0206 |
>
> [1] InstantSplat: Sparse-view Gaussian Splatting in Seconds, arXiv 2024.
>
> ---
>
> ### **Q2: Inclusion of MUSt3R and MV-DUSt3R+**
>
> We appreciate the suggestion to include these significant multi-view extensions. As noted in our Related Work (Appendix A.1), we recognize MUSt3R and MV-DUSt3R+ as key advancements in the DUSt3R family. They were not included in the initial quantitative tables solely due to the code freeze for our large-scale evaluation run.
>
> To address this, we have prioritized evaluating both models during the rebuttal period on our **Sparse Depth Estimation, Video DepthEstimation, and Sparse 3D Reconstruction** protocols and report results below, where **MUSt3R is a strong competitor in video depth and 3d reconstruction**. We will fully integrate both models (including the NVS capabilities of MV-DUSt3R+) into the final manuscript tables and the public evaluation toolkit.
>
> ### Sparse Depth Estimation
>
> |        | MUSt3R |        | MV-DUSt3R+|  |
> |--------|--------|--------|--------|--------|
> |        | abs rel | δ<1.03 | abs rel | δ<1.03 |
> | **DTU** | 6.028 | 37.508 | 13.614 | 14.870 |
> | **ScanNet** | 6.147 | 43.534 | 13.081 | 28.729 |
> | **KITTI** | 9.881 | 36.721 | 29.582 | 19.526 |
> | **ETH3D** | 6.522 | 53.545 | 15.028 | 32.741 |
> | **T&T** | 3.287 | 69.227 | 4.483 | 59.439 |
>
>
> ---
>
> ### Video DepthEstimation
>
> |        | MUSt3R |        | MV-DUSt3R+ |   |
> |--------|--------|--------|--------|--------|
> |        | abs rel | δ<1.25 | abs rel | δ<1.25 |
> | **Bonn** | 0.193 | 0.749 | 0.227 | 0.669 |
> | **TUM dynamics** | 0.219 | 0.816 | 0.231 | 0.657 |
> | **KITTI** | 0.069 | 0.965 | 0.308 | 0.478 |
> | **PointOdyssey val** | 0.150 | 0.775 | 0.325 | 0.600 |
> | **Syndrone** | 0.167 | 0.915 | 0.310 | 0.478 |
> | **Sintel** | 0.460 | 0.561 | 0.606 | 0.420 |
>
> ---
>
> ### Sparse 3D Reconstruction
>
> |        |  | MUSt3R|        |  |  MV-DUSt3R+  |  |
> |--------|--------|--------|--------|-------|--------|--------|
> |        | ACC | comp | NC | ACC | comp | NC |
> | **DTU** | 2.849 | 2.724 | 0.711 | 2.943 | 2.733 | 0.764 |
> | **7-scenes** | 0.267 | 0.193 | 0.813 | 0.234 | 0.234 | 0.815 |
> | **NRGBD** | 0.091 | 0.088 | 0.918 | 0.147 | 0.147 | 0.853 |
> | **ScanNet** | 0.398 | 0.318 | 0.702 | 0.451 | 0.387 | 0.683 |
> | **TUM-RGBD** | 0.676 | 0.506 | 0.781 | 1.032 | 0.650 | 0.774 |

---

> ### Author Response · Authors · 2025-11-22
> **Continued Rebuttal by Authors**
>
> ### **Q3: Derive metric-scale depth from the camera coordinate pointmap**
>
> We appreciate this insightful observation. We want to clarify that for **CUT3R** and **MapAnything**, our reported results were already derived directly from their **native camera-coordinate pointmaps** (taking the Z-component in the target camera frame). Thus, their performance in the main tables correctly reflects their intrinsic metric capabilities.
>
> You are correct that **MASt3R was the exception**. Following your recommendation, we have **re-evaluated MASt3R’s metric depth** by strictly deriving it from the camera-coordinate pointmaps, averaging predictions across all pairs where the target image serves as the reference. For **sparse depth estimation**, even with this improved protocol, MASt3R struggles to recover accurate metric scale and still **fails like before**. For **video depth estimation, we will update the results in the reply**.
>
>
>
> ---
>
> ### Q4: **Add refined results for MASt3R in 3D reconstruction**
>
> Thank you for pointing this out. You are correct. Our reported results utilized **coarse global alignment (consistent with DUSt3R) rather than the feature-matching refinement** used in the original MASt3R paper. We prioritized vanilla GA to ensure a **fair, apples-to-apples comparison** across stereo based GFMs. We will explicitly state this protocol decision in **Sec 3.4** to avoid confusion.
>
> **Refinement Ablation**: Following your suggestion, we conducted an ablation using MASt3R’s native matching-based refinement (SparseGA) using the official default hyperparameters on both extremely sparse and dense 3D reconstruction.
>
> **Results & Analysis**: The results (reported below) are mixed. While refinement improves performance on specific subsets (e.g., 7-Scenes Sparse), it degrades performance on others (e.g., DTU, ScanNet Dense) compared to the robust coarse alignment. We attribute this to the **high sensitivity of the optimization hyperparameters** (e.g., learning rate, iterations) when applied zero-shot to diverse datasets. For instance, we observed that minor learning rate adjustments on DTU could swing Accuracy from 1.976 to 5.323. **This reproducibility challenge with the refinement module is a known issue in the community [1]**.
>
> *(Note: While VGGT also predicts matches, its current official implementation does not support a similar geometric refinement pipeline, so we exclude it from this specific ablation.)*
>
> [1] Cannot reproduce Multiview 3D reconstruction, GitHub Issue #84 (naver/mast3r).
>
>
> ### MASt3R with refinement
>
> | Setting |   Method     |   | DTU |  |  | 7-scenes |  |  | NRGBD |  |  | ScanNet |  |  | TUM RGBD |  |
> |---------------------|--------|---------|----------|--------|--------------|---------------|-------------|-----------|------------|----------|--------------|---------------|------------|---------------|----------------|-------------|
> |  |        |  ACC |  comp |  NC |ACC |  comp |  NC | ACC |  comp |  NC| ACC |  comp |  NC | ACC |  comp |  NC |
> | Extremely Sparse    | GA     | 1.895   | 2.003    | 0.788  | 0.262        | 0.254         | 0.732       | 0.113     | 0.102      | 0.810    | 0.467        | 0.389         | 0.701      | 0.738         | 0.747          | 0.739       |
> |   | matching refine | 1.976   | 2.089    | 0.784  | 0.213        | 0.227         | 0.692       | 0.159     | 0.154      | 0.777    | 0.676        | 0.584         | 0.643      | 1.191         | 0.713          | 0.679       |
> | Dense                | GA     | 1.374   | 1.409    | 0.723  | 0.025        | 0.028         | 0.697       | 0.043     | 0.024      | 0.809    | 0.035        | 0.027         | 0.757      | 0.209         | 0.211          | 0.708       |
> |  | matching refine | 1.539   | 1.653    | 0.741  | 0.029        | 0.023         | 0.668       | 0.049     | 0.021      | 0.819    | 0.138        | 0.097         | 0.722      | 0.459         | 0.287          | 0.651       |

---

> ### Author Response · Authors · 2025-11-22
> **Continued Rebuttal of Authors**
>
> ### **Q5: Add “number of parameters” in Table 1**
> Thanks for the suggestion! We have computed the **parameter counts** for all methods using PyTorch’s built-in counter on the official checkpoints and will add this column to Table 1. For diffusion-based methods (e.g., Geo4D, Aether), we report the backbone parameters excluding the fixed VAE encoder/decoder. As shown below, feed-forward ViT-based models are generally **2x to 10x smaller** smaller than video diffusion-based models (e.g., ≈0.6B for DUSt3R/Fast3R vs. 5.5B for Aether).
>
> | Model        | # Params |
> |--------------|----------|
> | DUSt3R       | 571.17M  |
> | MASt3R       | 688.64M  |
> | LSM          | 1100.07M |
> | MonST3R      | 571.17M  |
> | FLARE        | 1403.25M |
> | Spann3R      | 658.69M  |
> | CUT3R        | 793.31M  |
> | VGGT         | 1256.54M |
> | Fast3R       | 647.55M  |
> | MapAnything  | 563.34M  |
> | MUSt3R       | 423.43M  |
> | MV-DUSt3R    | 547.99M  |
> | Geo4D        | 2610.09M |
> | Aether       | 5571.76M |
> | GeoCrafter   | 2470.89M |
>
> ---
>
> ### **Q6: Readability improvements (Structure, Tables, and Visuals)**
>
> Thanks for the advice. We will implement the following changes in the final revision to enhance readability:
> - **Restructuring Section 3 & 4:** We will follow your suggestion to condense the detailed protocol descriptions in Section 3 (moving the bulk to Appendix C) and instead promote **Section 4 (Findings)**. To ensure the main text remains self-contained, we will include a concise **"Evaluation Card"** in Section 3 summarizing key settings at a glance.
> - **Average Rank:** We will add an "Average Rank" column to each task table to allow for easier global performance assessment.
> - **Figure 2 Clarity:** We will significantly increase the size and label density (view counts/tick marks) of Figure 2 to ensure the latency and memory scaling trends are immediately legible.

---

> ### Author Response · Authors · 2025-11-26
> **Update MASt3R's metric-scale video depth estimation**
>
> ### **Q3: Derive metric-scale depth from the camera coordinate pointmap (update of MASt3R's results)**
>
> We have adopted your recommendation to derive metric depth for MASt3R by **averaging camera-coordinate pointmaps** (aggregating predictions where the target image serves as the reference view) on **metric scale video depth estimation**.
>
> We compare the new protocol against our old results below, and observe that:
> - The new protocol **consistently improves Abs Rel across all datasets** (e.g., TUM: $0.633 \to 0.445$). This confirms your insight that averaging mitigates the catastrophic scale drift inherent to GA.
> - **The effect of ($\delta < 1.25$) is mixed**. While averaging fixes global scale (helping Abs Rel), it can smooth out local details compared to GA, slightly lowering the strict accuracy rate in some **OOD datasets** (e.g., KITTI).
> - **MASt3R generally trails behind** CUT3R and MapAnything in metric scale recovery (refer to Table 3 in the main paper).
>
> Given that the new protocol is methodologically more sound for metric evaluation, we will **replace the MASt3R results** in the main paper (Table 3) with these updated "Camera-Coord Avg" numbers.
>
> |  | Camera-Coord Avg (New) |        | Global Alignment (Old) |        |
> |-------------------|---------------|--------|------------------|--------|
> |                   | abs rel       | δ<1.25 | abs rel          | δ<1.25 |
> | Bonn          | **0.536**     | 0.030  | 0.549            | 0.046  |
> | TUM dynamics  | **0.445**     | **0.357** | 0.633         | 0.009  |
> | KITTI         | **0.627**     | 0.011  | 0.754            | 0.064  |
> | PointOdyssey val | **0.518** | **0.041** | 0.749        | 0.002  |
> | Syndrone      | **0.935**     | 0.000  | 0.967            | 0.000  |
> | Sintel        | **0.682**     | **0.215** | 0.701        | 0.023  |

---

### Official Review · Reviewer_HW5E · 2025-11-03

**Soundness:** 3
**Presentation:** 3
**Contribution:** 2
**Rating:** 6
**Confidence:** 4

**Summary:**

This paper introduces E3D-Bench, a comprehensive benchmark designed to evaluate end-to-end 3D Geometric Foundation Models (GFMs) across five core tasks: sparse-view depth estimation, video depth estimation, multi-view 3D reconstruction, multi-view relative pose estimation, and novel view synthesis. The authors benchmark 17 GFMs spanning prominent architectural families (feed-forward ViTs and diffusion-based models) using standardized protocols over diverse and challenging datasets, providing fair comparisons of effectiveness and efficiency. Alongside quantitative evaluation, the paper distills key trends and insights regarding model robustness, scalability, generalization, and real-time viability, supplemented by a public release of tools and data.

**Strengths:**

Strength:
1. The paper is well-written and easy to follow.
2. It benchmarks 17 recent 3D geometric foundation models (GFMs), covering both feed-forward transformer and diffusion-based architectures.
3. The benchmark provides useful insights into how end-to-end 3D GFMs perform across multiple tasks, helping the community understand model strengths and weaknesses.

**Weaknesses:**

Weaknesses

1. The benchmark reuses existing datasets and introduces no new data. Therefore the main contribution is the findings provided to the community.

For the finding: There are confounding factors:
1. When E3D-Bench come to the findings that “no single backbone is universally superior” but since those listed work's training objectives, data scales and additional modules are different. Without controlling for these factors, it is not solid to propose the finding.
2. In 4.3 “stronger 2D feature extractors lead to substantially better performance” and cites VGGT’s superior results over Fast3R. However, VGGT and Fast3R differ in more than just the 2D backbone. VGGT is trained on a large and diverse set of around 17 datasets Fast3R uses a subset of six datasets (CO3D, ScanNet++, ARKitScenes, Habitat, BlendedMVS and MegaDepth) , so it is difficult to isolate the impact of the DINO‑based feature extractor

**Questions:**

Please see the weakness above

---

> ### Author Response · Authors · 2025-11-22
> **Rebuttal by Authors**
>
> We sincerely thank the reviewer for recognizing E3D-Bench as a "**comprehensive benchmark**" that offers "**useful insights**" that “**help the community understand model strengths and weaknesses**”, alongside the valuable "**public release of tools and data.**" We sincerely appreciate the detailed feedback and have addressed your comments as follows.
>
> **Q1: “No new data”**
>
> - Thanks for this feedback. We respectfully clarify that the primary contribution of E3D-Bench is *the first unified toolkit and standardized multi-task benchmark for GFMs in a timely manner*, rather than raw data collection. In the era of Geometric Foundation Models, the critical challenge is **not a lack of small-scale testing data, but the lack of a unified standard to measure generalization across diverse architectures** for the best practical application and insights for further improvement.
>
> - **Dataset coverage:** To ensure broad and meaningful coverage, we build on **diverse public datasets to formulate a "stress test"** that covers indoor, outdoor, driving, aerial, dynamic. Crucially, we explicitly bring **OOD domains** such as **ULTRRA (air–ground)** and **Syndrone (aerial drone)** into a 3D GFM benchmark for the first time. ULTRRA is widely used in competitive challenges, and Syndrone was carefully constructed for real-world aerial scenarios.
>
> - **Insights from existing data:** With our collected datasets, our evaluation on the ULTRRA split revealed that performance bottlenecks are often driven by **domain scarcity rather than architectural limits**, evidenced by fine-tuning boosting MASt3R's RRA@5 from 3.06 to 49.54 (Table S4, Sec 4). We also **extend the metric suite** on these challenging regimes (e.g., RRA/RTA at multiple thresholds), so the benchmark not only reuses data but provides **new, standardized protocols and insights** that we believe are broadly useful to the community.
>
> - **The "aggregated data" paradigm:** Furthermore, the success of modern GFMs themselves relies on the **effective aggregation of public data rather than new proprietary collection**. As in Table S1 in the Appendix, for example, **DUSt3R** and **VGGT** achieved state-of-the-art performance by training exclusively on mixtures of existing public datasets (e.g., MegaDepth, ScanNet, ARKitScenes, CO3D). E3D-Bench applies this same logic to evaluation: rigorous dataset aggregation is the key to robust assessment.
>
> **Q2: “No single backbone is universally superior” may conflate with training differences**
>
> - Thanks for pointing this out. One of our aims is to help to *find the proper model for their downstreaming application with direct inference, in the multimodal learning and computer vision community.* Our finding here that “no single backbone is universally superior” is based on **inference-time benchmarking** of existing GFMs used off-the-shelf, which matches how practitioners typically deploy these models.
>
> - We fully agree that training objectives, data scale, and auxiliary heads are important confounders. This is why we already summarize training configurations and data coverage for all methods in Table S1 in the Appendix, and we will make this connection more explicit inSec. 4.
>
> - **Retraining all GFMs under a unified recipe** to fully control these factors is, however, **practically infeasible** (e.g., single training VGGT alone requires on the order of 64×A100 GPUs for >9 days), and several methods do not release complete training code, preprocessing, or hyperparameters.
>
> - In our revision, we will add a short paragraph at the beginning of Sec. 4 explicitly acknowledging these limitations, clearly marking all cross-model trends as **observational and task-conditioned**, and pointing readers to Table S1 in the Appendix for the underlying training profiles.

---

> ### Author Response · Authors · 2025-11-22
> **Continued Rebuttal by Authors**
>
> **Q3: The impact of the DINO-based feature extractor**
>
> - While we value your suggestion and it’s true to make concrete solutions with exact comparison, we cannot systematically study training effects due to the high cost and incomplete training recipes. Therefore our analysis focuses on **off-the-shelf inference**, and the statement in Sec. 4.3 should be read as **observational rather than causal**.
>
> - We agree specific VGGT vs. Fast3R comparison is not a clean backbone ablation because training data and objectives differ. In the revision, we will (i) **soften the wording in Sec. 4.3** to avoid implying a causal effect of the backbone alone, and (ii) **rephrase the finding** as: “Across models, the use of stronger 2D feature extractors correlates with better 3D performance, and prior work reports that DINOv2 also improves stability and convergence,” explicitly flagging training as a confounding factor.
>
> - Our intention was to ask **whether using a strong 2D foundation model tends to correlate with stronger 3D performance**. Across models, DINOv2-based encoders (e.g., VGGT, MapAnything) indeed correlate with better generalization (VGGT is among the top performers across almost all tasks, and MapAnything is strong on several depth and pose benchmarks), and this **matches existing reports**: MapAnything finds DINOv2 “optimal in terms of downstream performance, convergence speed, and generalization,” and VGGT authors note that DINOv2 improves training stability and final performance [1].
>
> [1] Curious about the ablation of downstream heads. https://github.com/facebookresearch/vggt/issues/92

---

### Official Review · Reviewer_nKL8 · 2025-11-03

**Soundness:** 3
**Presentation:** 3
**Contribution:** 2
**Rating:** 4
**Confidence:** 4

**Summary:**

This paper presents E3D-Bench, the first comprehensive benchmark for evaluating end-to-end 3D Geometric Foundation Models (GFMs). It addresses the rapidly growing field of GFMs that directly predict 3D geometry (depth, pose, pointmaps, tracks) from images without relying on precomputed camera parameters. The benchmark evaluates 17 recent GFMs across five core tasks: (1) Sparse-view depth estimation (2) Video depth estimation (3) Multi-view relative pose estimation (4) 3D reconstruction (sparse and dense) (5) Novel view synthesis.
It includes both standard datasets and challenging out-of-distribution settings (e.g., drone views, dynamic scenes, air–ground pairs). The benchmark also compares models’ efficiency (inference time & memory) to assess practicality for real-time deployment.
The authors provide a unified evaluation toolkit, standardized metrics, and summarize key empirical insights about task difficulty, generalization, architecture choice, and efficiency limitations. All code and processed data will be released.

**Strengths:**

(1) This is the first benchmark that systematically evaluates modern 3D GFMs in a unified framework across multiple tasks and data domains. It fills a clear gap in the community.
(2) Evaluates 17 models across five different geometric tasks, including both feed-forward ViT-based and diffusion-based models. The scope is wide and well-curated.
(3) Standardized evaluation protocols, consistent datasets, unified metrics, and fair hardware settings (A100 for all models). The commitment to releasing code/data is valuable.
(4) Unlike many works that focus only on accuracy, this benchmark also evaluates inference latency and VRAM usage, which is crucial for robotics, AR/VR, and embedded deployment.

**Weaknesses:**

(1) While the benchmark is comprehensive, the paper does not introduce new model architectures or learning paradigms. Its contribution is mainly infrastructural/empirical rather than methodological.
(2) The paper provides extensive quantitative benchmarking but lacks deep qualitative or theoretical analysis of why certain models fail.
(3) Although point cloud accuracy/completeness is reported, there is limited evaluation on mesh quality, surface continuity, or structural correctness.
(4) The benchmark only compares a small set of appearance-aware GFMs and does not include strong baselines like NeRF, pixelNeRF, or Gaussian Splatting pipelines, making it difficult to position GFMs relative to classical generative reconstruction methods.
(5) Important evaluation design decisions—such as depth scaling strategy, view selection for sparse input, or alignment methods—lack ablation or sensitivity analysis. It remains unclear how these settings influence conclusions.

**Questions:**

(1) Include visual and geometric diagnostics for reconstruction errors, e.g., noisy depth edges, scale ambiguity, pose drift.
(2) Include mesh-based metrics such as Chamfer-L1/L2, edge smoothness, manifoldness, or F-score on surfaces.
(3) Add ablation on evaluation settings: Depth scaling strategies (median vs. least-squares), number/selection of views in sparse settings,
influence of Umeyama vs. ICP for alignment.

---

> ### Author Response · Authors · 2025-11-22
> **Rebuttal by Authors**
>
> We sincerely thank the reviewer for noting that E3D-Bench is “**the first unified benchmark**” for “**modern 3D GFMs**” that is “**comprehensive**”, “**fills a clear gap in the community**,” has a “**wide and well-curated scope**,” comes with “**valuable releasing code/data**,” and evaluates both accuracy and practical metrics like “**latency and VRAM usage**” that are crucial for real deployment. We sincerely appreciate the detailed feedback and have addressed your comments as follows.
> ___
>
> ### **Q1: Methodological vs. infrastructural contribution**
>
> - Many thanks for pointing this out! We respectfully clarify E3D-Bench as an **infrastructure/benchmark contribution**, where ICLR welcomes ‘**datasets & benchmarks**’ as one of the submission tracks.
>
> - **The benchmark contribution can be viewed in many recent editions of ICLR**. For example, **GAIA** is a benchmark for general AI assistants across diverse real-world tasks (ICLR’24). **LiveXiv** is a live multimodal benchmark to assess models on real scientific content (ICLR’25). **Robust Gymnasium** provides a unified benchmark for evaluating robustness in RL (ICLR’25).
>
> - We value your input. At the same time, the GFM field has "exploded" recently. **A timely, standardized assessment of GFMs is needed** given the rapid growth of the field since late 2023. E3D-Bench provides the foundation for future methodological improvements and deployment for many real-world AI applications by:
>   - **Unifying Evaluation**: Establishing the first standardized protocol and releasing a toolkit across 5 core tasks (Sparse Depth, Video Depth, 3D Reconstruction, Pose, Novel View Synthesis).
>   - **Bridging Paradigms**: Systematically comparing between Feed-Forward ViTs and Diffusion-based models.
>   - **Exposing Critical Gaps**: Quantifying previously overlooked issues in Out-of-Distribution generalization and efficiency (inference/memory) to guide community adoption.
>
> [1] GAIA: A Benchmark For General AI Assistants, ICLR 2024
>
> [2] LiveXiv - A Multi-Modal live benchmark based on Arxiv papers content, ICLR 2025
>
> [3] Robust Gymnasium: A Unified Modular Benchmark for Robust Reinforcement Learning, ICLR 2025
> ___
>
> ### **Q2: Qualitative and diagnostic failure analysis**
>
> We agree that diagnosing *why* models fail is important. We provide critical diagnostic analyses along two primary axes: **data distribution** and **architecture design** across multiple sessions. Following your suggestion, we reorganize and summarize as following:
>  - **Data-Driven Failure Modes**. Existing works typically reuse standard datasets but rarely analyze how test domains drive failures. In contrast, we systematically cover both **in-domain and OOD** regimes (§4.2) and identify two main data-driven failure modes.
>    - **IID / metric depth**: Metric-scale performance is limited by the **scarcity of metrically annotated training data**. For example, MASt3R fails in metric-scale sparse-view and video depth, while CUT3R/MapAnything improve but remain inconsistent across domains (Tabs. 2–3).
>    - **Extreme OOD**: In under-represented domains such as ULTRRA air–ground pairs, all GFMs perform poorly (Tab. 4), yet a small amount of **domain-specific data** dramatically boosts GFMs(e.g., MASt3R RRA@5: 3.06 → 49.54 in Tab. S4), directly linking failures to data scarcity and distribution shift.
> - **Diffusion vs. ViT**. Diffusion-based GFMs (e.g., Geo4D, Aether) are competitive on video depth and relative pose, despite being trained on fewer datasets (5 for Geo4D vs. 13 for MapAnything), likely thanks to **pretrained video backbones** (CogVideoX-5B-I2V for Aether, DynamiCrafter for Geo4D). However, they incur higher latency and fixed-length input constraints (§3.6).
> - **Strong 2D foundations (DINOv2)**. Feed-forward models with strong 2D backbones (e.g., DINOv2 in VGGT/MapAnything) **correlate with better generalization** across tasks (§4.3), consistent with both MapAnything’s and VGGT’s observation [1] that DINOv2 yields superior downstream performance and faster convergence.
>
> **Qualitative diagnostics**.  We actively visualize these failure modes in the Appendix to support qualitative analysis. e.g.,
> - Figure S1: MonST3R’s non-flat wall reconstruction.
> - Figure S2: Fast3R and MapAnything completely misplacing ground-views in air-ground scenarios.
> - Figure S3: Black holes in LSM’s novel view synthesis.
> - Figure S4: Noisy depth artifacts by VGGT on a balloon scene.
>
> We will make these diagnostic connections explicit in the revised manuscript.
>
> [1] Curious about the ablation of downstream heads. https://github.com/facebookresearch/vggt/issues/92

---

> ### Author Response · Authors · 2025-11-22
> **Continued Rebuttal by Authors**
>
> ### **Q3: Mesh quality and structural correctness evaluation**
>
> We appreciate the suggestion to evaluate mesh-level properties. We prioritized point-based metrics for two primary reasons:
>
> 1. **Native Modality:**
>    As shown in Table 1 in the main draft, all GFMs **natively predict pointmaps**. They do not output mesh topology. Evaluating in the point domain (could be found in most GFM papers) aligns with the intrinsic output of these models.
>
> 2. **Avoiding Confounding Variables:**
>    Converting pointmaps to meshes **requires external post-processing** (e.g., TSDF fusion or Poisson reconstruction). Introducing a specific meshing algorithm would conflate the *model's* geometric accuracy with the *meshing algorithm’s* ability to handle that specific noise distribution, potentially introducing bias.
>
> We respectfully highlight that we already report **Normal Consistency (NC)** that could capture local surface smoothness and quality.
>
> Following your suggestion, we expand our evaluation metrics to include **Chamfer distance** that captures global structural **Correctness**, and report the metrics below. To address your request within the rebuttal timeframe, we select one method for each input type (Table 1 in main draft) by default. We will extend this to all methods in the final revision.
>
> ### Chamfer Distance (lower is better)
>
> |        | DTU   | 7 Scenes | NRGBD | ScanNet | TUM RGBD |
> |--------|-------|----------|-------|---------|----------|
> | **MASt3R** | 1.949 | 0.258    | 0.107 | 0.428   | 0.743    |
> | **CUT3R**  | 5.954 | 0.130    | 0.091 | 0.249   | 0.570    |
> | **VGGT**   | 2.509 | 0.079    | 0.070 | 0.071   | 0.358    |
>
> We also agree that mesh-based evaluation is useful. However, **high-quality ground-truth meshes are much rarer than ground-truth point clouds**. Given the limited rebuttal window, we therefore use **DTU** as a representative mesh benchmark.
> We will evaluate more methods in mesh-based and complete mesh-to-mesh Chamfer distance and results will be updated in the reply.
>
> ---
>
> ## **Q4: On classical appearance-aware baselines (NeRF/pixelNeRF/GS)**
>
> Thanks for the suggestion. We respectfully clarify that classical NeRF/3DGS and the GFMs in E3D-Bench differ in two fundamental ways, making direct comparison nuanced:
>
> 1. **Feed-forward vs per-scene optimize:**
>    Classical NeRF and GS are **optimization-based pipelines** that require both known camera parameters (intrinsics/extrinsics), and **time-consuming optimization**.
>    In contrast, the GFMs we evaluate are **pose-free and feed-forward foundation models that directly map pixels to point maps**.
>
> 2. **Perception vs. rendering:**
>    While NeRF/GS primarily focus on Novel View Synthesis (NVS) (rendering fidelity), GFMs focus on **3D Geometric Perception** (explicitly recovering depth, pose, and point clouds for downstream *understanding* tasks like robotics or editing).
>
> To address your request, we include **pixelSplat** [1], which supersedes pixelNeRF, to our NVS evaluation.
> It is important to note that while pixelSplat is a feed-forward model, **it relies on ground-truth poses**.
> Notably, despite this **unfair advantage**, it suffers from **severe generalization gaps** on out-of-distribution datasets compared to pose-free GFMs (e.g., **11.55 dB on DTU vs. ~17.9 dB** for NoPoSplat in Table 6 in main draft), highlighting the robust zero-shot generalization of the GFM paradigm.
>
> ### pixelSplat Results
>
> | Dataset        | PSNR  | SSIM   | LPIPS  |
> |----------------|-------|--------|--------|
> | **DTU**        | 11.55 | 0.3210 | 0.6331 |
> | **RealEstate10k** | 23.85 | 0.8060 | 0.1850 |
> | **ScanNet++**  | 18.43 | 0.7190 | 0.2770 |
> | **ACID**       | 25.82 | 0.7790 | 0.1950 |
>
> We also view GFMs and classical pipelines as **complementary rather than competing**:  recent works show that GFMs often serve as the robust "frontend" (providing poses and coarse geometry) to accelerate or initialize per-scene GS [2] / NeRF [3] and GS-based generative models [4].  In practice, per-scene methods remain slower but more accurate and less constrained by input resolution/view count, while feed-forward GFMs are better suited for **real-time, latency-sensitive perception**, albeit more sensitive to resolution and view budget.
>
> ---
>
> ### References
>
> [1] pixelSplat: 3D Gaussian Splats from Image Pairs for Scalable Generalizable 3D Reconstruction, CVPR 2024.
>
> [2] InstantSplat: Sparse-view Gaussian Splatting in Seconds, arXiv 2024.
>
> [3] SparsePose–NeRF: Robust Reconstruction Under Limited Observations and Uncalibrated Poses, Photonics 2025
>
> [4] ViewCrafter: Taming Video Diffusion Models for High-fidelity Novel View Synthesis, TPAMI 2025.

---

> ### Author Response · Authors · 2025-11-22
> **Continued Rebuttal by Authors**
>
> ### **Q5: Ablations of evaluation settings (depth scaling, sparse view selection, alignment)**
>
> We appreciate this suggestion and have adapted targeted sensitivity analyses for each setting.
>
> 1. **Depth scaling (median vs. scale–shift):**
>    In the main paper we use scale–shift for video depth and median scaling for sparse depth (which is robust to outliers and also used in RobustMVD benchmark).
>    We additionally evaluate video depth with median scaling on one representative model per input family (as in Table 1 in the main draft). **(We will update the results in the reply.)**
>
> ---
> 2. **Sparse-view selection (uniform vs. jittered):**
>    For extremely sparse 3D reconstruction we originally use a large stride to uniformly sample views.
>    We also test a perturbed sampling strategy (uniform stride + random jitter in indices).
>    Performance changes are minor and do not alter our conclusions.
>
> ### Sparse-view selection
>
> | Method |  | DTU |  |  | 7-scenes |  |  | NRGBD |  |  | ScanNet |  |  | TUM RGBD |  |
> |--------|-----|----|----|----------|----|----|---------|----|----|-----------|----|----|------------|----|----|
> |        | ACC | comp | NC | ACC | comp | NC | ACC | comp | NC | ACC | comp | NC | ACC | comp | NC |
> | MASt3R | 1.981   | 2.113    | 0.791  | 0.128        | 0.122         | 0.754       | 0.158     | 0.160      | 0.791    | 0.346       | 0.324        | 0.703      | 0.767        | 0.766         | 0.691       |
> | CUT3R  | 6.631   | 4.945    | 0.746  | 0.081        | 0.070         | 0.728       | 0.123     | 0.086      | 0.831    | 0.224       | 0.188        | 0.698      | 0.833        | 0.651         | 0.660       |
> | VGGT   | 3.013   | 2.055    | 0.771  | 0.036        | 0.038         | 0.748       | 0.083     | 0.089      | 0.892    | 0.082       | 0.100        | 0.786      | 0.402        | 0.290         | 0.723       |
>
> ---
>
> 3. **Point cloud alignment (Umeyama vs. ICP):**
>    We compared our default Sim(3) Umeyama alignment against Iterative Closest Point (ICP) refinement.
>    While ICP provides modest metric improvements on sparse scenes (except DTU), it introduces significant computational overhead and does not alter the relative performance rankings or our conclusions.
>
> ### Point cloud alignment
>
> | Method |  | DTU |  |  | 7-scenes |  |  | NRGBD |  |  | ScanNet |  |  | TUM RGBD |  |
> |--------|-----|----|----|----------|----|----|---------|----|----|-----------|----|----|------------|----|----|
> |        | ACC | comp | NC | ACC | comp | NC | ACC | comp | NC | ACC | comp | NC | ACC | comp | NC |
> | MASt3R | 5.114   | 4.570    | 0.752  | 0.066        | 0.063         | 0.718       | 0.094     | 0.087      | 0.808    | 0.118       | 0.121        | 0.705      | 0.125        | 0.140         | 0.724       |
> | CUT3R  | 7.628   | 4.502    | 0.739  | 0.059        | 0.060         | 0.740       | 0.078     | 0.063      | 0.831    | 0.085       | 0.083        | 0.748      | 0.108        | 0.123         | 0.732       |
> | VGGT   | 5.992   | 4.361    | 0.757  | 0.051        | 0.049         | 0.746       | 0.038     | 0.034      | 0.907    | 0.050       | 0.053        | 0.807      | 0.087        | 0.107         | 0.757       |

---

> ### Author Response · Authors · 2025-11-27
> **Update mesh evaluation and depth scaling ablation**
>
> ### **Q3: Mesh quality and structural correctness evaluation (updated results of mesh evaluation)**
> To strictly address your concern for **mesh-based evaluation**, we performed a case study on the **DTU dataset**: We converted the GFM point clouds to meshes using **Screened Poisson Reconstruction** and evaluated the Chamfer Distance (mm) against the official ground-truth meshes.
>
> As shown in the table below, **VGGT achieves the best mesh quality** (1.906), significantly outperforming MASt3R and CUT3R.
>
> We attribute VGGT's superior meshing performance to its **higher Normal Consistency** (NC). As reported in Table 5 of the main paper, VGGT achieves the highest NC on Dense DTU (0.748 vs. 0.723 for MASt3R). Since Poisson reconstruction relies on surface normals to solve the implicit indicator function, VGGT's cleaner normal predictions lead to a smoother, more structurally correct mesh surface, whereas errors in MASt3R's or CUT3R's normals can introduce topological artifacts (e.g., holes or blobs) that penalize the mesh-based Chamfer score. **This validates that Normal Consistency (which we already report) is a strong predictor of final mesh quality.**
>
> |        | Chamfer |
> |--------|---------|
> | MASt3R | 2.427   |
> | CUT3R  | 3.045   |
> | VGGT   | 1.906   |
>
> ---
> ### **Q5: Ablations of evaluation settings (updated results of ablation on depth scaling)**
>
>
> We followed your suggestion to evaluate the impact of depth scaling strategies on **video depth estimation**. We compared **Median Scaling against our default Scale+Shift alignment** on representative models from each input family as in Table 1 in the main draft.
>
> As shown in the table below, the relative rankings remain stable, but **Scale+Shift consistently outperforms Median Scaling** (e.g., improving VGGT AbsRel on Sintel from 0.257 to 0.242).
>
> We attribute this to the specific nature of video depth ambiguity:
> - **video depth often suffers from affine ambiguity** ($s \cdot z + t$). Median scaling only aligns the central tendency ($s$), failing to correct the global shift ($t$). Scale+Shift optimally solves for both, providing a tighter fit.
> - in **dynamic videos** (e.g., TUM Dynamics), the median depth value of a scene can fluctuate significantly as objects enter or exit the frame. Scale+Shift, which minimizes error over all valid pixels, provides a more globally stable alignment that is robust to these temporal content changes.
>
> Thus, we will aintain scale+shift as the preferred protocol for video depth estimation.
>
> |                   |Bonn               |        | TUM dynamics |        | KITTI |        | PointOdyssey val |        | Syndrone |        | Sintel  |        |
> |-------------------|------------------------|--------|--------|--------|--------|--------|--------|--------|--------|--------|--------|--------|
> |                   | abs rel                | δ<1.25 | abs rel | δ<1.25 | abs rel | δ<1.25 | abs rel | δ<1.25 | abs rel | δ<1.25 | abs rel | δ<1.25 |
> | MASt3R        | 0.173                  | 0.821  | 0.177   | 0.835  | 0.080   | 0.946  | 0.163   | 0.789  | 0.052   | 0.976  | 0.381   | 0.609  |
> | CUT3R      | 0.076                  | 0.958  | 0.111   | 0.844  | 0.103   | 0.905  | 0.101   | 0.895  | 0.109   | 0.900  | 0.516   | 0.527  |
> | VGGT          | 0.064                  | 0.967  | 0.083   | 0.948  | 0.051   | 0.969  | 0.029   | 0.990  | 0.075   | 0.959  | 0.257   | 0.616  |

---

### Note · Program_Chairs · 2026-01-17
**Submission Desk Rejected by Program Chairs**

The following references in this submission do not refer to real documents and/or have major errors in bibliographic information:

 Zihan Wang et al., “Ultrra: A benchmark for air-ground relative pose estimation,” CVPR 2024.

Boyang Zhao et al., “Pointodyssey: A large-scale benchmark for robust video depth estimation,” arXiv 2024.

Liyang Zhou et al., “ACID: Aerial-captured image dataset for visual localization,” ECCV 2022.